# Millimeter-wave to near-terahertz sensors based on reversible insulator-to-metal transition in VO$_2$

Fatemeh Qaderi[1✉], Teodor Rosca[1], Maurizio Burla[2], Juerg Leuthold[2], Denis Flandre[3] & Adrian M. Ionescu [1✉]

In the quest for low power bio-inspired spiking sensors, functional oxides like vanadium dioxide are expected to enable future energy efficient sensing. Here, we report uncooled millimeter-wave spiking detectors based on the sensitivity of insulator-to-metal transition threshold voltage to the incident wave. The detection concept is demonstrated through actuation of biased VO$_2$ switches encapsulated in a pair of coupled antennas by interrupting coplanar waveguides for broadband measurements, on silicon substrates. Ultimately, we propose an electromagnetic-wave-sensitive voltage-controlled spike generator based on VO$_2$ switches in an astable spiking circuit. The fabricated sensors show responsivities of around 66.3 MHz.W$^{-1}$ at 1 μW, with a low noise equivalent power of 5 nW.Hz$^{-0.5}$ at room temperature, for a footprint of $2.5 \times 10^{-5}$ mm$^2$. The responsivity in static characterizations is 76 kV.W$^{-1}$. Based on experimental statistical data measured on robust fabricated devices, we discuss stochastic behavior and noise limits of VO$_2$ -based spiking sensors applicable for wave power sensing in mm-wave and sub-terahertz range.

[1] Nanoelectronic devices laboratory (Nanolab), Department of Electrical Engineering, École polytechnique fédérale de Lausanne (EPFL), EPFL STI IEL NANOLAB, ELB 335, Station 11, Lausanne 1015, Switzerland. [2] Institute of Electromagnetic Fields (IEF), Eidgenössische Technische Hochschule Zürich (ETHZ), ETZ K 82, Gloriastrasse 35, Zürich 8092, Switzerland. [3] ICTEAM, Ecole Polytechnique de Louvain (UCLouvain), ELEN, Place du Levant 3/L5.03.02, Louvain-la-Neuve 1348, Belgium. ✉email: fatemeh.qaderi.92@gmail.com; adrian.ionescu@epfl.ch

Vanadium dioxide ($VO_2$) is a transition metal oxide with correlated electrons which undergoes a first-order insulator-to-metal transition (IMT) at 340 K from a low-temperature monoclinic insulating state to a high-temperature rutile metallic state, accompanied by structural, electrical, and optical changes[1,2]. The potential of IMT along with its sensitivity to external stimuli makes it promising for a variety of applications in resistive memories[3–5], optical switches[6,7], sensors[8–10], tunable photonic devices[11,12], brain-inspired and neuromorphic architectures[13–15]. Moreover, RF/MW applications of tunable structures are reported based on various types of $VO_2$ layers[16,17]. We target the sensor application in the frequency range of mm-wave and terahertz (THz). THz detectors are widely growing in different applications in spectroscopy[18], astronomy[19], biomedicine[20], imaging[21], surveillance security[22], high-data-rate communication[23], etc.

Aside from thermal stimuli, the phase transition can be triggered by other perturbations, such as optical pumping or changes in electric field, pressure, and doping. The mechanism of electric-field-driven IMT has been under debate for a long time. Formation of conducting filaments at bias voltages above a threshold is explained in some studies[24], while the mechanism of filamentation has been differently interpreted, such as the non-uniform current distribution followed by local Joule heating[24–26], or a breakdown mechanism similar to the one in thin oxides[27]. On the other hand, it has been discussed that the electric field can induce IMT through nucleation of conducting filaments similar to the mechanism in chalcogenide-based phase-change memory and switches[28] or without the need for a thermal process in Mott insulators[29–31]. The physics of such transitions is still under debate but it is recognized that it involves nucleation processes related to conductive path formation that are stochastic in nature[32].

Resistive switching in $VO_2$ based on electric-field-assisted carrier generation, which leads to critical doping levels making the Mott insulator unstable, allows moderate fields to excite a large number of carriers with negligible heating[33]. The idea of electric-field-driven carrier generation is also reinforced by localized triggering of IMT by means of carbon nanotube (CNT)s to enhance the electric field locally and decreasing the required voltage for IMT in $VO_2$ thin films[34]. Also, the feasibility of ultrafast $VO_2$ switching into a metastable metallic state[35] indicates a tunneling mechanism, in which long-lived metallic domains are generated by intense multi-terahertz excitation. The idea of above-thermal-process speed is also reinforced by THz spectroscopy of the ultrafast photoinduced IMT in $VO_2$[36].

Considering the thermal requirements for transition between insulating ($M_1$) and metallic (R) crystal phases ($2.32\,eV/nm^3$) calculated by integrating heat capacity and latent heat[2], the fundamental process of photoinduced phase transition turns out to show a different mechanism from a complete thermally induced transition, and ultimately a density-driven mechanism is suggested for this transition[37]. This is essentially practical in the case of THz and mm-wave, as they cannot actuate carriers through the band-gap in a semiconductor. A first demonstration of field-induced IMT of VO2 was reported using picosecond THz waveforms[38].

Among room temperature technologies for millimeter-wave (mm-wave)/ THz detection, one can consider field effect transistor (FET) detectors[39–47], Graphene or 2D material-based structures[48–55], diodes and rectifiers[56–58], photoconductive detectors[59,60], and bolometers[61,62]. The sensing mechanism in FET detectors is generally through the direct voltage induced in the channel according to the plasma waves[63]. 2D material-based transistors and switches mostly detect based on the transfer function of the channel which is activated by radiation[52].

Photoconductive materials are used in detectors according to their photo-responsivity (activation of the carriers by exposure)[60]. Diodes and rectifiers have a big role in THz detection for their nonlinearity-based, fast response. The two latter mechanisms are also used in heterodyne systems. Bolometers range from different materials which show a temperature coefficient of resistivity (TCR) and, particularly because $VO_2$ has a considerable TCR, it is worth comparing the $VO_2$-based detectors with our device. We have summarized most of the above detectors compared with our proposed one in terms of responsivity, noise, and speed in the discussion section.

There are specific challenges in detection mechanisms based on $VO_2$ thermal response such as $VO_x$ micro-bolometers, which use TCR in these materials[64,65]. Experimental $VO_2$-based bolometers[66–69] have reported the operation by exploiting the semiconductor phase of $VO_2$, i.e., at room temperature or normal temperatures well below the phase change (where $TCR_{normal} = 2–6\%K^{-1}$[69]). The cliff of the transition is not used experimentally, however, there are simulation[65] and modeling works[70,71] assuming $TCR_{max}$ upto $171\%K^{-1}$[65]. Moreover, according to the experimental data of the transition edge bolometer in[66], operating in the hysteresis region does not lead to any expected higher responsivity, but even half of the one when operated at $TCR_{normal}$ in periodic acquisition mode, due to formation of minor hysteresis loops and the loop accommodation process[66]. The high-resistance $VO_2$ does not have a noticeable absorption of THz at the operating temperature of these bolometers[72]. This motivates the common use of THz absorbers for THz sensing by bolometers.

In our method, instead of using the slope of resistance, we monitor the location of the cliff and its displacements due to the incoming signal. The modulation of this transition with the signal is converted into a variation of the oscillation frequency as our detection scheme is a $VO_2$-based stochastic oscillator. The observed IMT voltage modulation was interpreted as field-induced enhancement of carrier density (Poole–Frenkel effect) succeeded by Joule heating due to electron-lattice coupling, which then stabilizes the metallic phase. For transition edge $VO_2$ bolometers, the lattice instabilities close to the phase-change temperature along with the local Joule heating can interfere with the bias control, as well as confining the dynamic range of the input[71,73]. In the device configuration of our work we are no more confronted by such instabilities and the power range can go broader (over a milli-watt) compared to that of bolometers (up to 20 μW[64]).

We design and experimentally demonstrate a stochastic power sensor by investigating the interaction of mm-wave and $VO_2$ at a controlled ambient under small DC voltage bias (below 2.5 V) at room temperature. The responsivity is at least three times larger than the similar sensors[74], along with a limit of detection (LoD) comparable to the state of the art[75], and a footprint of one order of magnitude smaller than the typical of $VO_2$ sensors[74].

We report on the stochastic behavior of switching based on experimental statistical data measured on robust fabricated $VO_2$ devices, using a spike oscillator readout. We harness the stochasticity, which translate in Poisson noise challenges. In order to study $VO_2$ electrical properties and the material interaction with the electromagnetic wave within a broadband of mm-wave range, we conduct 2-port experiments on a coplanar waveguide (CPW) interrupted by a $VO_2$ device in the signal line. The experiments are generally based on the actuation of $VO_2$ by mm-wave, assisted by a collective effect of a bias voltage. Using a pair of coupled antennas, one as the emitter and the other as the receiver, we establish the wireless short-range detection concept in mm-wave range. The receiving antenna has been designed and fabricated with $VO_2$ encapsulated in the area where it concentrates the incoming wave.

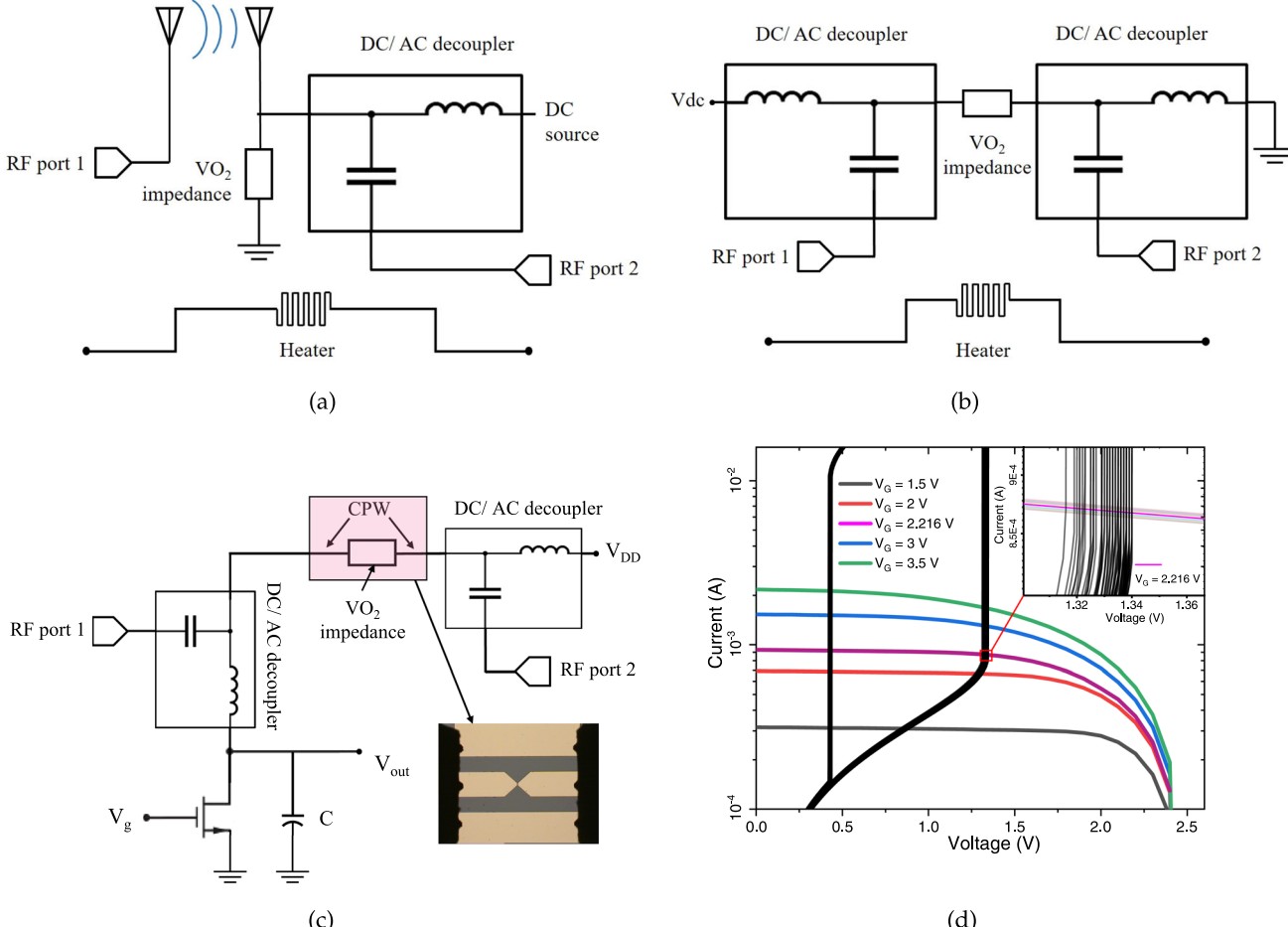

**Fig. 1 Mm-wave detection method and the experimental setup. a** VO$_2$ in coupled antennas case, **b** VO$_2$ in interrupted CPW case, **c** astable circuit using a 2-terminal VO$_2$ device in interrupted CPW in series with a MOSFET, **d** the initial operation point of the oscillating sensor is set by the intersection of the static characteristics of VO$_2$ ($V_{VO_2}$)and the series drain-source voltage of the NMOS transistor ($V_{DD} - V_{ds}$) at specific points supporting astability. Inset: the operational bias points and multiple cycles of experiments suggesting non-deterministic behavior.

According to the available facilities the antennas are designed for mm-wave frequency range, while further scaling the antennas design (which is feasible through the same fabrication process for dimensions down to 1 µm) can shift the operational frequency up to THz. In terms of the detection mechanism, the material shows responsivity to a wide range of input frequencies from RF to THz, infrared (IR), visible, and UV[76–78]. Ultimately, we propose a readout circuit by means of an astable multivibrator formed of a VO$_2$ impedance in series with an NMOS transistor (Fig. 1c), in which the change in the frequency of oscillations represents the power applied to the VO$_2$ sensor.

## Results

### Principles of detection using VO$_2$ IMT threshold voltage shifting

It is recognized that different excitation sources applied simultaneously can create a collective effect to assist the phase transition in correlated oxides such as VO$_2$. According to the measured static current-voltage characteristics of VO$_2$ 2-terminal switches, the IMT threshold voltage shows a decline at elevated temperatures[79]. This gives the inspiration of combining a DC voltage with an external stimulus to implement the core of a sensing device. We exploit the DC-bias voltage along with the mm-wave signal to influence the IMT/ MIT transition points. The physics of this phenomenon involve modulating the formation of conductive filaments across the thin VO$_2$ film. The contribution

of electric fields in filament formation has been suggested by several works[24–26].

The electromagnetic power sensor is based on a two-terminal VO$_2$ device on which the mm-wave radiation is concentrated by an adapted antenna design (Fig. 1a), or, by directly applying a controlled power of high-frequency signal to the device under test in a dedicated setup (Fig. 1b).

By investigation of two-port S parameters and the static $I$–$V$ characteristics of the VO$_2$ switches, we probe the transition behavior in presence of an excitation signal at different power levels.

A voltage bias enhances the VO$_2$ sensitivity to the incoming wave and the antennas give enhancement to the absorption. A general schematic of the setup for antenna measurements is illustrated in Fig. 1a, with high-frequency signal and the bias voltage de-coupled.

For broadband measurements, we used a CPW with the signal line interrupted by a VO$_2$ device. After the proof of concept by the antennas, in order to study the VO$_2$ response to the mm-wave power with a higher resolution, we conducted static measurements on CPW-based devices. The simplified equivalent schematic for CPW experiments is shown in Fig. 1b.

An efficient sensing configuration is proposed as depicted in Fig. 1c, by connecting the 2-terminal VO$_2$ device in series with a conventional MOSFET and load a capacitor in their middle point, in order to build an astable oscillator. The two alternating output

voltages of this circuit correspond the two threshold voltages of the VO$_2$ switch. The frequency of oscillations in the output of the circuit in Fig. 1c can be modulated by the power injected through the RF ports due to the modulation of the VO$_2$ threshold voltage.

According to Fig. 1d, the astability conditions are fulfilled at specific bias points where the transistor's current intersects that of the VO$_2$ impedance immediately after IMT threshold, as experimentally demonstrated in Supplementary materials note 1: Operational range of the spike generator. The horizontal axis is $V_{VO_2} = V_{DD} - V_{ds}$, where $V_{DD}$ is the supply voltage at 2.5 V, and $V_{ds}$ is the voltage between the drain and the source of the transistor in Fig. 1c.

The mm-wave signal is generated by multiplier modules connected to the vector network analyzer (VNA) and then directed to the transmission lines through GSG infinity probes. To maintain the experiments reproducible, in all the configurations a heater (as a temperature reservoir) is connected to the chuck to keep the sample at a fixed temperature. More details are found in Supplementary materials note 2: Setup Considerations.

**VO$_2$ actuation through coupled antennas**. By design and fabrication of a pair of coupled antennas, we established an experimental scheme to evaluate the VO$_2$ actuation through electromagnetic radiation power. At the receiver antenna, a 2-terminal VO$_2$ device is encapsulated inside the antenna where the incoming wave is concentrated. A cross section of the middle part inside the receiving antenna as well as a microscopic image are included in Fig. 2a, b, respectively.

The VO$_2$ middle volume (lateral size: $2.5 \times 10^{-5}$ mm$^2$) is regarded as the sensitive area of the device. The antennas of Fig. 2b are 350 μm in length, 20 μm in width, and 80 nm in thickness and have a resonance frequency around 145 GHz as shown in the return loss of Fig. 2c.

The VO$_2$ polycrystalline grains are revealed in the zoomed parts of Fig. 2b by scaning electron microscopy (SEM). The closer we bias the receiving antenna to the transition voltage, the more sensitive the IMT becomes to the radiation. The incident wave will add up to the threshold and actuate the metallic phase in VO$_2$.

A radiation between 130 GHz and 160 GHz is applied to the emitter antenna to cover the range within which the coupling between the two antennas is at a peak. The emitter antenna is identical to the receiver by the metallic pattern, while it has no VO$_2$ and is not biased.

The S parameters over frequencies ranging from 40 to 220 GHz are shown in Fig. 2c, d. When VO$_2$ is in insulating phase, the receiver antenna is terminated by the port from VNA with a standard impedance. Therefore, the impedance matching at this phase is at its highest, and so does the transmission through the antennas. As soon as VO$_2$ goes metallic, the antenna gets shunted by the smaller resistance of VO$_2$. Therefore, the value of $|S_{21}|$ gets suppressed by the change in the antenna impedance at the receiver. The receiver port matching and therefore the resonance in $|S_{11}|$ disappears as well, due to the dominant low impedance of VO$_2$. A theoretical simulation of the S parameters by finite element method (FEM) for different values of VO$_2$ conductivity is also in agreement with these results, provided in Supplementary materials note 3: Antennas Considerations.

Having the maximum $|S_{21}|$ between the antennas ($-11$ dB), the lower bound for insertion loss from the VNA to the VO$_2$ part of the antenna becomes 22 dB, excluding other potential power loss (see "Methods" section). At room temperature, even the maximum incident power to the VO$_2$ antenna provided by the VNA is not sufficient to cause a transition, as illustrated in Fig. 2c, d; However, the presence of a bias voltage splits the curves dramatically by VO$_2$ undergoing IMT due to radiation.

In the DC characteristics of the VO$_2$ sample of Fig. 2b, the IMT occurs near 1.713 V. The power level of 0 dBm at the VNA output (i.e. an effective power level of $-22$ dBm at the receiver), can split the curves for VO$_2$ biased at any voltage higher than 1.690 V. Closer to IMT critical voltage, for example at 1.695 V, the minimum power required to make the transition reduces to $-10$ dBm at VNA output (effectively $-32$ dBm at the receiver). Right after the event of IMT, we set the VNA power back to $-30$ dBm, which previously could not trigger a transition, but now the curves remain at the metallic state values. This is in complete agreement with the hysteretic characteristics of the transition. It is worth noting that by removal of the bias voltage, the backward transition (MIT) occurs immediately.

The DC characteristics of the VO$_2$ devices inside the antenna are depicted in Fig. 2e for different levels of mm-wave power. The distribution of these threshold voltages is found in Fig. 2f.

The equivalent RMS AC voltage amplitude corresponding to the mm-wave power that can trigger the IMT in Fig. 2c, d is smaller than the difference between the bias voltage and the threshold voltage. Reciprocally, the DC shift in the IMT threshold voltage of Fig. 2e is greater than the equivalent RMS AC voltage amplitude of the corresponding mm-wave power. A similar behavior is observed when linking the RMS AC signal to an equivalent DC voltage in a nonlinear system, for instance, when operating RF micro-electromechanical systems (MEMS) capacitive switches biased near the pull-in voltage[80].

**VO$_2$ actuation through interrupted CPW**. For a wide-band study on the VO$_2$-based detector, we fabricated CPWs with the signal line interrupted by a VO$_2$ device. Directly applying the mm-wave power through the VNA ports connected to both ends of a CPW, we try to minimize possible transmission losses, in order to extract the effect of mm-wave exposure on VO$_2$ DC characteristics. Using AC/DC decouplers, voltage-controlled I(V) current limited measurements were performed on the two terminals of the interrupted CPW. The CPW along with the RF probes are shown in Fig. 3a.

To characterize the transfer function of S parameters versus VO$_2$ state, we measured the S parameters at several temperatures from 25 °C up to 90 °C, having VO$_2$ through semiconducting, mixed states and metallic state. Figure 3b shows the average magnitude of S parameters for the CPW over frequencies from 60 GHz up to 110 GHz in intermediate states of VO$_2$ from semiconducting to metallic. Despite the relatively sharp IMT by voltage actuation, VO$_2$ transition has a tail in resistance versus temperature characteristics for a few Celsius degrees, as well as an observable slope. The evolution of the curves at higher temperatures is quite consistent with the gradual expansion of metallic domains in VO$_2$ by temperature[32]. The frequency response over this range is provided in Supplementary materials note 4: CPW S parameters.

We measured the static $I$–$V$ characteristics of VO$_2$ inside the CPW in presence of a wide-band signal ranging from above mm-wave (60 GHz) up to 110 GHz. The curves reveal a considerable shift in the static IMT critical voltage of VO$_2$. At each power, 100 double-sweeps of voltage are recorded and plotted for each of Figs. 4a–c, at room temperature, 40°, and 50°, respectively. The curve-splitting gets smaller at temperatures closer to $T_{IMT}$ as expected, besides the additional thermal fluctuations which reduce the resolution between the curves at different power levels. The extracted probability densities of the IMT threshold voltages are depicted in Figs. 4d–f for each power level. Most of the distributions cannot be fitted with normal Gaussian distribution curves and need a bimodal fitting that could relate to nonlinear phenomena specific to stochastic systems like correlated oxides.

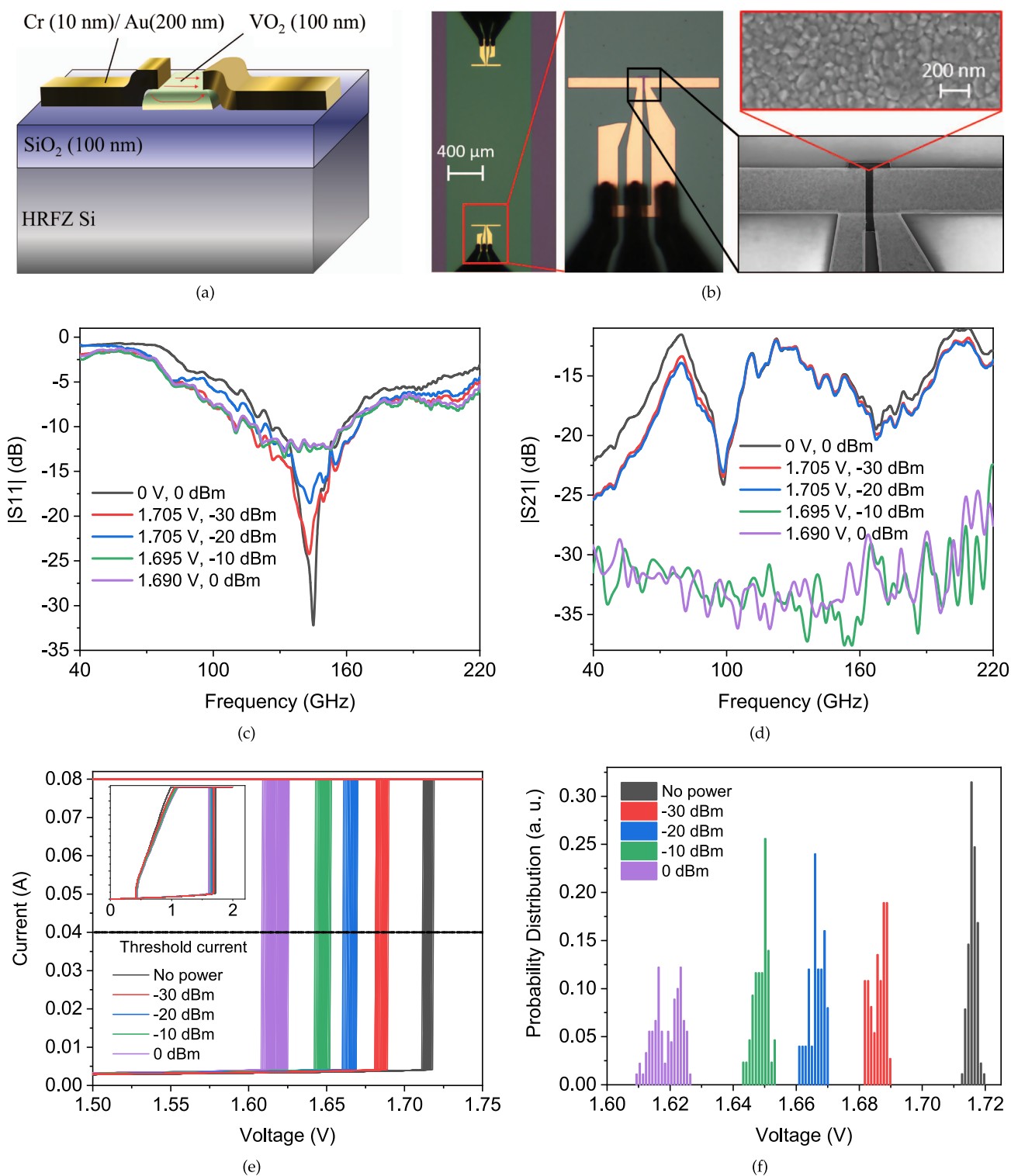

**Fig. 2 Coupled dipole antennas structure and performance. a** Cross section of the central part in the receiver antenna (Note: the emitting antenna has no VO$_2$ area), **b** microscopic image of the receiver antenna along with the zoomed SEM image, **c** |$S_{11}$| for the emitting port for different bias voltages: impedance mismatch trend representing the transition in VO$_2$ shunting the receiver, **d** |$S_{21}$| for the coupled antennas at different bias voltages: splitting curves in "on" and "off" states according to the impedance transition, **e** multicycle measurement of I–V characteristics for VO$_2$ in the receiving antenna, showing the shift in the IMT threshold voltage according to the external power. Inset: an overview of the whole characteristics, **f** probability density of IMT threshold voltages at each power.

**Astable circuit**. The 2-terminal VO$_2$ devices function as active elements in the astable multivibrator circuit of Fig. 1c. The output signal of the oscillator was recorded for several thousands of cycles at each power level. We notice that the average of the IMT threshold voltage ($V_{IMT}$) sensitivity to the incident power is much higher than that of the average MIT threshold voltage ($V_{MIT}$). Thus, we obtain a modulation of the hysteresis window width as a function of applied external RF power.

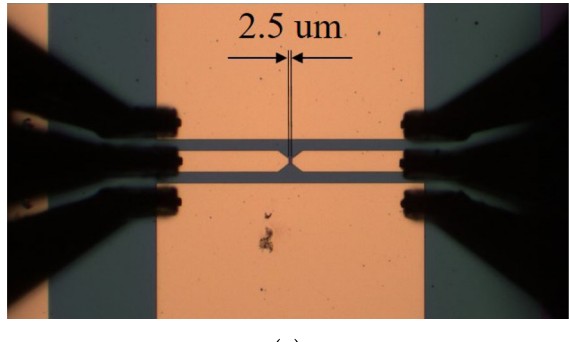

(a)

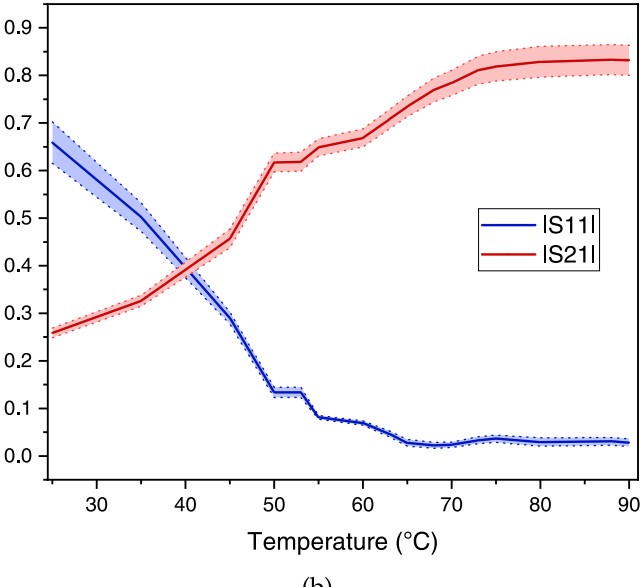

(b)

**Fig. 3 Interrupted CPW by VO$_2$ structure and S parameters behavior.**
**a** Under the microscope, **b** S parameters magnitude between the two ports across the CPW vs. temperature. The standard deviation is represented as the shades around the curves.

Any change in the IMT threshold voltage modulates the intersection of the VO$_2$ device I–V characteristics with that of the transistor in a series configuration (Fig. 1d). As long as the excursion of the operating point does not reach the triode region of the MOSFET, any change in hysteresis window width will translate into a proportional change in operating frequency, as shown in the fast Fourier transform (FFT) spectra of Fig. 5a. The FFT is performed using a rectangular window over an interval of 3 s.

The histograms at each power level (Fig. 5c–h) show bimodal distributions which can be related to the threshold voltage distributions of Fig. 4d–f through the nonlinear oscillator transfer function. Every waveform is recorded for about 3 s and every cycle is converted to an equivalent frequency to be compared statistically.

Based on Allan deviation calculations, we were able to identify four main noise behaviors. One related to thermal noise, leads to a Normal statistic and dominates for averaging times below 200 ms. The lowest noise level around 200 ms is usually related to 1/f (Flicker) noise. The next rise in the distribution noise can be related to the nucleation Poissonian processes and dominates for longer time constants. Final bumps are mostly due to parasitic signals, inter-modulation, low-frequency interference, instabilities of supply voltage, etc. It then limits the potential averaging of

successive frequency measurements to reduce their standard deviation. Both mechanisms combine to induce threshold fluctuations, act as precursors of a bifurcation, and appear to form the basis of the stochastic oscillating behavior.

**Detection capability.** The shift in the static IMT critical voltage in presence of a high-frequency signal, with respect to the baseline static curve (the one with no additional high-frequency power), is a criterion for the "response" of the VO$_2$-based devices. Using a proper read-out circuit, one can exploit the sensing capability of such a response in mm-wave and THz applications. Responsivity, or $R_V$, defined as the ratio of the shift in the required voltage for IMT to the radiation power from the emitting port, is plotted in Fig. 6a versus the corresponding power: $R_V = \frac{\Delta V}{P}$, where $\Delta V$ is the shift in the static threshold voltage regarding the applied power ($P$). The responsivity in Fig. 6a is based on the nominal power from the source, while if we consider only the received power by the VO$_2$ sensing area, the responsivity gets at least one order of magnitude higher.

We apply a similar concept for responsivity in the frequency shift of oscillations, defined by the deviation in the central frequency of oscillations divided by the corresponding power: $R_f = \frac{\Delta f}{P}$, where $\Delta f$ is the shift in the frequency of oscillations with respect to the reference frequency. The reference frequency is attributed to the free running oscillation under no applied power (Fig. 5a).

The noise equivalent power (NEP) of a detector is theoretically defined as the ratio between the noise power spectral density and the responsivity. One source of the noise in this detector is the VO$_2$ resistance and therefore responsible for Johnson-Nyquist noise, which depends on temperature and resistance by $\sqrt{4k_B T R \Delta f}$, where $R$ is the resistance of the active detecting area and $\Delta f$ is the measurement single-sided bandwidth. Based on the insulating phase, where the resistance of the device is maximum 1.4 kΩ, the upper bound for the Johnson-Nyquist noise is evaluated 4.8 nV.Hz$^{-1/2}$. This is a simplified estimation of the noise spectral density. Another source is the shot noise of the bias current. The root mean square (RMS) current fluctuations as a Poisson process has a magnitude of $\sqrt{2qI_{dc}\Delta f}$, where $q$ is the charge of an electron and $I_{dc}$ is the bias current. In order to experimentally evaluate the noise in static measurements, we apply a constant voltage close to $V_{IMT}$ and record the fluctuations in the current over 200 s, and calculate the NEP down to 4 mHz through the FFT of the signal fluctuations (Fig. 6b). In an alternative measurement, we fixed the current and measured the voltage fluctuations. The results for both show relatively the same levels of noise, which we attribute more to external sources and the setup.

The procedure of extracting the NEP: We normalize the FFT of the current-fluctuations-voltage-product by the bandwidth and extract the integrated output noise power ($P_{N,out}$) through the analysis described in the Supplementary materials note 5: NEP evaluation. The noise spectral density (NSD) voltage would be equal to $v_n = \sqrt{P_{N,out}.R_L}$.

Where $R_L$ is the measurement tool impedance, typically 50 Ω. The equivalent input noise power ($P_{N,in}$) is then evaluated by normalizing $v_n$ by the voltage responsivity: $P_{N,in} = v_n/R_V$. The responsivity value used here is the one at the smallest power detected. Finally, to achieve the NEP we divide this power by the $\sqrt{BW}$. For the case of static detection, a value of 1 Hz is set as the BW.

The maximum NEP among all the cases happens when the temperature goes high enough so that the degradation in responsivity dominates the decline in resistance, which explains

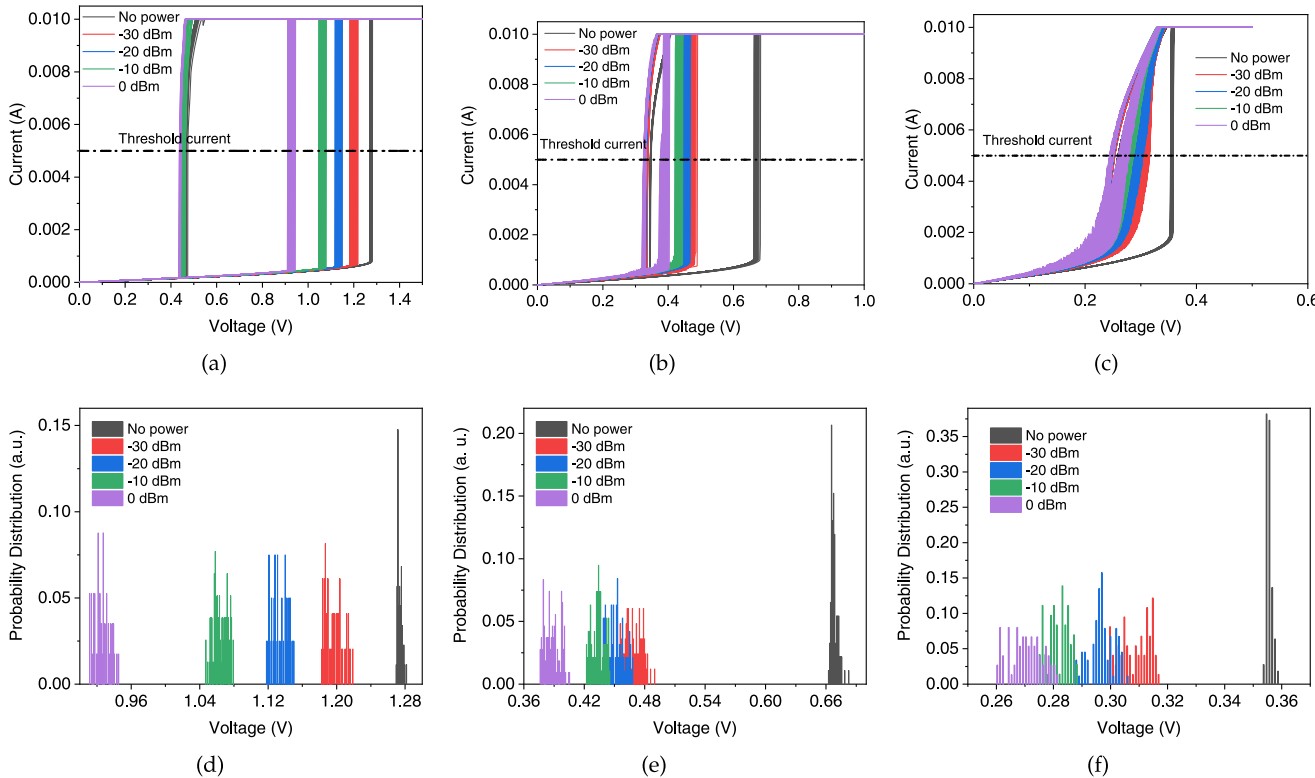

**Fig. 4 IMT threshold voltage distribution with respect to the mm-wave power. a–c** Static I–V characteristics of VO$_2$ with respect to different levels of external power, at **a** room temperature, **b** 40 °C, and **c** 50 °C. Probability distribution of IMT threshold voltage, at **d** room temperature, **e** 40 °C, and **f** 50 °C. Elevated temperatures make major shifts in the threshold voltage, while the effect of the external power is maintained in all cases. The temperature role is mainly on the resolution between the curves at different power levels.

the worst NEP in CPW devices at 50° C. At room temperature, we consider the worst NEP at low frequencies, which includes collectively the effect of material drifts, thermal fluctuations, flicker noise, and the external noise coming from the ambient setup. The upper bound of the NEP in this case is near 5 nW.Hz$^{-0.5}$ for frequencies higher than 1 Hz.

We also consider the intrinsic noise of the stochastic behavior in phase transition. The fitted standard deviation ($\sigma$) of ($V_{\text{IMT}}$) in CPW devices at room temperature, which is 2.77 mV when no power applied, determines the LoD. The LoD can be estimated by $3\sigma/R_V$ at lowest actuation level, which gives the value of 68.7 nW. However, the standard deviation for the frequency of oscillations according to Allan method[81] goes down to 1.3 Hz for averaging times around 200 ms when no power or very low power is applied (Fig. 6c). In this case the equivalent LoD will be 58 nW. The detailed LoD values at different power levels can be found in Supplementary materials note 6: Detailed LoD values.

Using the electrical responsivity, the minimum detectable signal (at SNR = 1) or the "electrical NEP", represents the intrinsic performance limitations of the device. While the optical NEP is equal to the ratio of the electrical NEP and the optical coupling efficiency of the detector system[82]. As we consider the emitted power by the VNA and not the absorbed one at the sensor spot, the evaluated responsivity and NEP are normalized by the coupling efficiency. Hence, we report an upper bound for the external NEP in this work.

**Response time**. In the experimental investigations based on application of electrical pulses, the IMT response time goes down to sub-nanosecond, and the MIT down to about one microsecond[83]. For direct measurements, every single acquisition will depend on the ramp generator speed as well as the probing instrument sampling rate. Every forward sweep in our case takes at least 200 ns to log a single phase transition (more explanations in Supplementary materials note 2: setup considerations).

For multicycle acquisition the reset time imposes a delay time of at least 1 μs. In the astable circuit, besides the material response time, the RC circuit time constant is also determining the timescales. It is 210 μs and 9 μs for "off" and "on" states, based on the 150 nF capacitor along with the effective resistance (VO$_2$ impedance in parallel with the transistor output resistance), while VO$_2$ impedances of 1.4 kΩ and 60 Ω for "off" and "on" states are dominant, respectively. It is worth noting that the experimental integration time in this case determines the duration of each acquisition. If we aim for the minimum NEP, we have to use more cycles of the same measurement, at least about 100 ms according to Allan method for this experiment, as in Fig. 6c.

We have investigated the timescales of partial switching through two-terminal VO$_2$ devices inside a CPW by applying a single-tone RF signal on one side and monitoring the same component on the other side by means of a spectrum analyzer. When no bias voltage is applied to the material, the output component is at the level of the noise floor by the spectrum analyzer. While in presence of a bias voltage right below the MIT threshold voltage, the component shows up as a demonstration of the material switching. The amplitude modulated THz measurements suggest a cut-off frequency (−3 dB) of 12 GHz for VO$_2$ partial switching in the hysteresis region of the material[84].

**Discussion**
The demonstration room temperature actuation of VO$_2$ by mm-wave and THz applied power proves the opportunity of designing and manufacturing efficient sensor devices based on the phase change phenomena. We first improved the sensitivity of VO$_2$ to

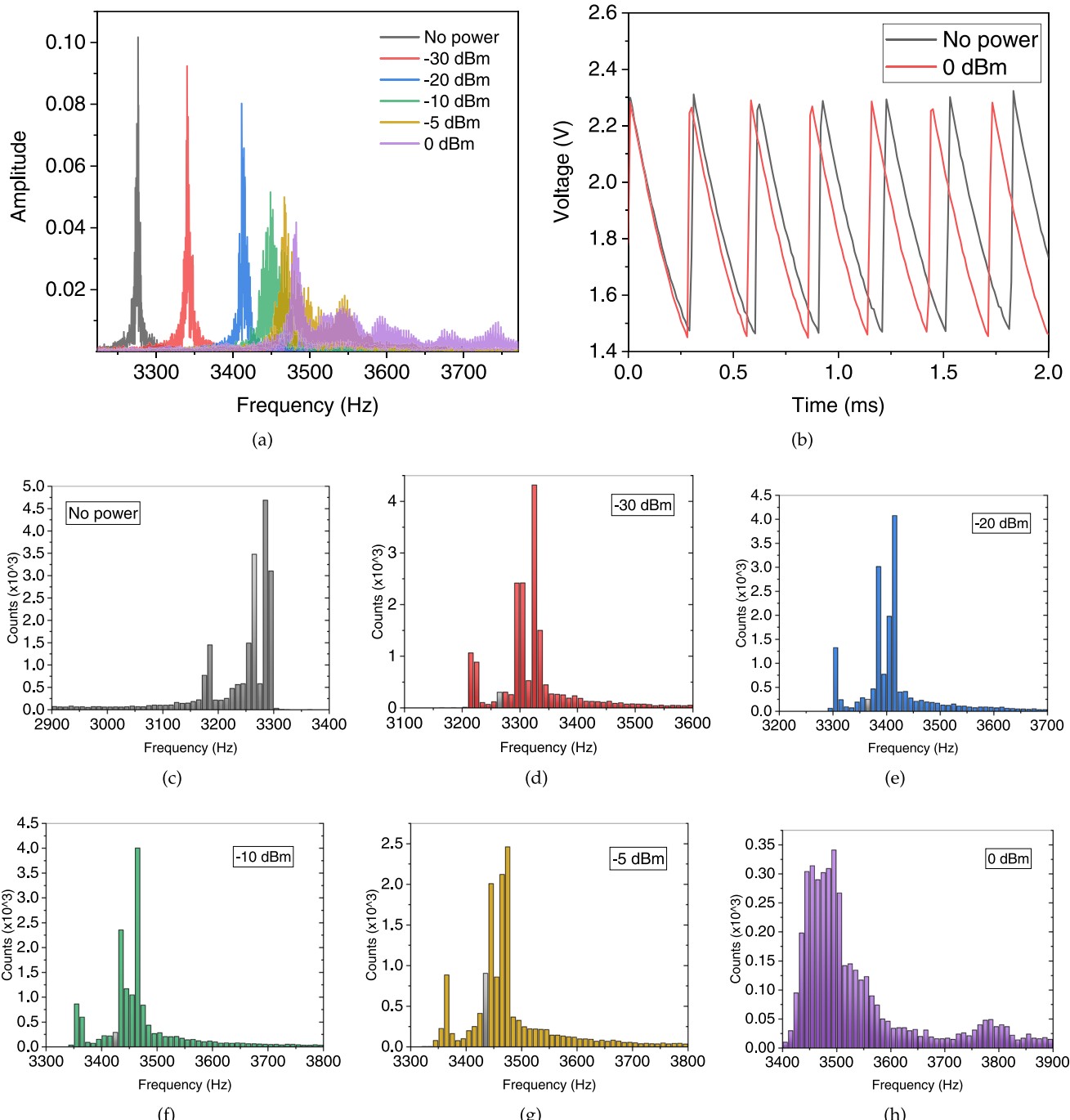

**Fig. 5 Distribution of the frequency of oscillations based on the mm-wave power. a** FFT of output waveforms with respect to the applied power, **b** time domain waveforms of the astable output signal for "no power" and "0 dBm" exposure, **c-h** the distribution of the frequency of oscillations based on the emitted power from the first port: every waveform is recorded for about 3 s and every cycle is converted to an equivalent frequency to be compared statistically.

low-power radiation by means of a DC-bias-assisted setting inside an antenna and investigated its properties in a CPW configuration.

We demonstrated the density-driven IMT by radiation in VO$_2$ through splitting the S parameters in a two-coupled-antenna setup, where VO$_2$ is embedded in one of them. Furthermore, the effect of external radiation on the static I-V characteristics of VO$_2$ is shown as a shift in the threshold voltage required to induce IMT. This effect can form the principle of detection in mm-wave and THz detectors. We reported the extent of this shift normalized by the radiation power as the nominal responsivity of the

device in Fig. 6a. This is while the actual responsivity as a function of absorbed power is at least 37 s higher than the values reported in this figure for the case of the antennas, and more than three times higher in the case of CPWs. With this definition, the equivalent noise power will also get lower proportionally.

The non-deterministic behavior of threshold voltage in VO$_2$ has roots in the nucleation processes responsible for the filament formation in a polycrystalline VO2 film. In each cycle of switching, the formation of paths can vary according to the local temperature and free electron distribution densities between the grains[85]. The switching voltage distribution can also be influenced

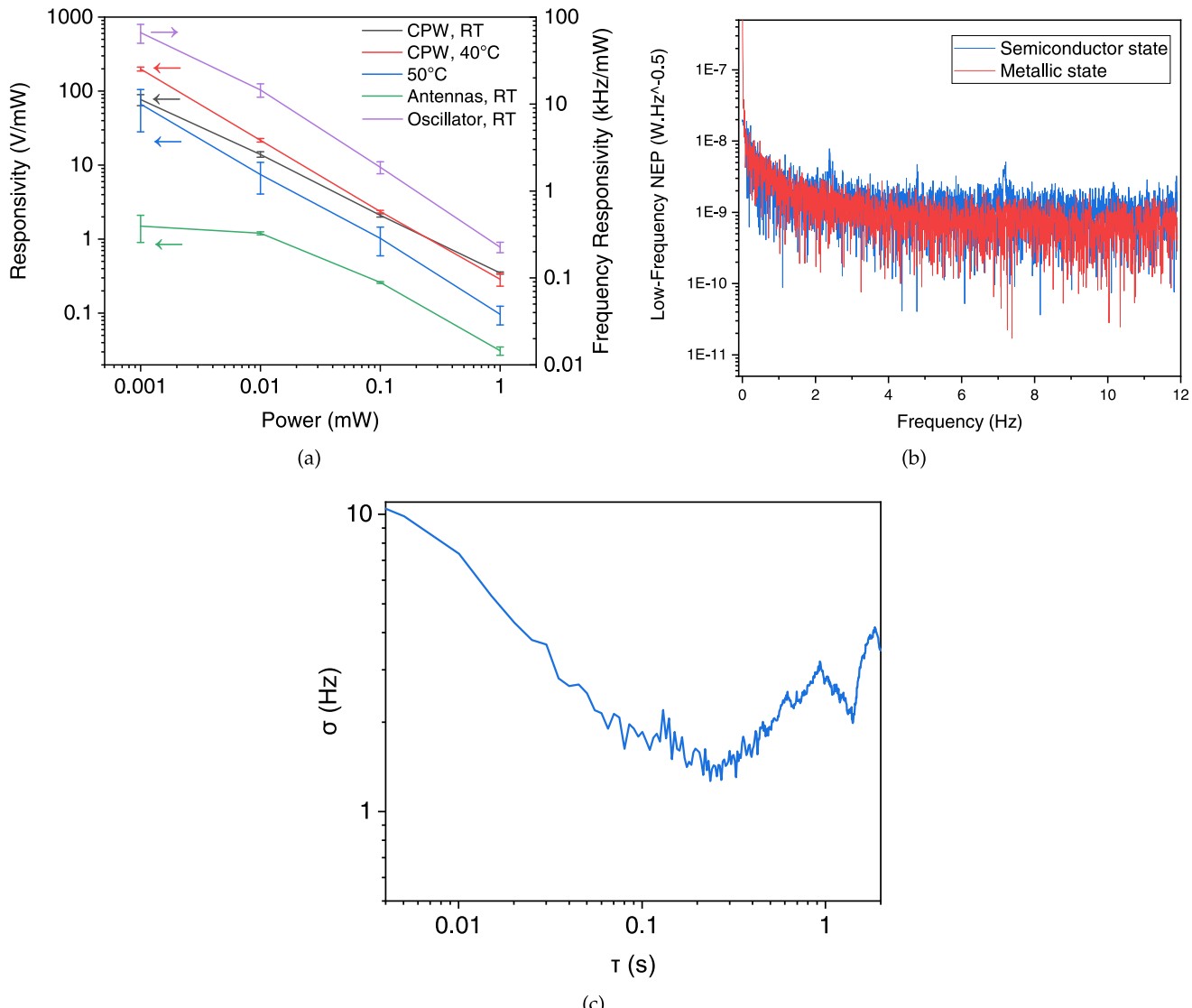

(a)

(b)

(c)

**Fig. 6 Performance figures of merit for the proposed VO₂-based sensor. a** DC and frequency responsivity due to the shift in the IMT threshold voltage for the mentioned configurations based on the emitted power from the first port. The error bars represent the standard deviation according to repetitive runs of measurements (at least 100 cycles for DC responsivity measurements and thousands of cycles for the oscillator response). **b** Noise equivalent power of the CPW devices at very low frequencies, at the minimum measured signal (1 μW), **c** Allan deviation calculation on the frequency vs time measurements of the oscillator over a long duration of time, no external power applied.

by non-stoichiometric parts of the film, such as variations in oxygen vacancies in different grains as in ref. [86]. A more general description for the non-stoichiometric effects are crystal defects. Grain boundaries also play a critical role in conduction and switching behavior[87]. Thanks to Allan deviation calculations, the uncertainty of power detection is reduced by almost one order of magnitude as inferred from Fig. 6c.

Concerning the nonlinear responsivity in Fig. 6a, the extent of the shift at the lowest incident power compared to the shift due to higher levels of power, confirms that the sensor shows a higher responsivity to smaller triggers. According to NEP = NSD/$R_V$, the higher responsivity at smaller excitations which are prone to a lower signal to noise ratio, allows more efficiency in low-signal detection. The equivalent RMS AC voltage is smaller than the actual DC shift that it causes. Such behavior is usually found in avalanche phenomena, and here it can relate to the lattice instabilities driving the phase transition at a proper bias point. Hence, by device scaling and bias point engineering, the figures of merit have room to be enhanced.

The reversible electric-field-induced hysteresis loop has shown robustness over millions of non-stop switching cycles in our oscillator measurements. The same devices have been measured over days, showing the same $V_{IMT}$ and $V_{MIT}$ threshold voltages. We inspected the degradation and aging of the VO₂ film regarding the switching and the ratio of resistance between the two states in Supplementary materials note 7: VO2 film investigation.

The response time of detectors based on a thermal process is usually longer than that of an electrical mechanism. The response time of our proposed sensor system is about 308 μs (mostly dictated by the readout instruments, not a VO₂ intrinsic characteristic), while it is about 83 ms for one of the latest state-of-the-art bolometers based on VO₂[68].

Regarding sensor applications, the above-described method and devices appear to be applicable over a relatively wide range in temperature from cooled conditions (the IMT voltage is always assisted by a calibrated DC voltage) up to around 40° C. The experimental figures of merit of the power sensor proposed here

**Table 1 Comparison of the detector performances in different technologies.**

| Detector type | $R_V$ | NEP [nW Hz$^{-0.5}$] | Frequency (THz) | Response time | Ref. |
|---|---|---|---|---|---|
| Graphene-based | 20–600 V W$^{-1}$ | 0.08–1 | 0.01–0.8/ 1.8–4.25 | 1 ns to 9 μs | 48,49,51–55 |
| Si FET | 10–5000 V W$^{-1}$ | 0.01–199 | 0.1–4.52 | 10 μs to 100 ms | 39–46 |
| Tunneling rectifier | 4 kV W$^{-1}$ | $2 \times 10^{-4}$ | 0.13 | NA | 56 |
| Schottky diode | 100–1000 V W$^{-1}$ | $4 \times 10^{-3}$–1 | 0.1–1 | 0.1 ns | 57,58 |
| Microbolometer | 500–760 V W$^{-1}$ | 0.1–0.2 | 1 | 140 μs | 61,62 |
| $^a$VO$_2$-based oscillator | 21.28 GHz/W | NA | Optical | 10 μs | 74 |
| VO$_2$-based bolometer | 36.9–124 V W$^{-1}$ | NA | 0.075–0.11 | 83 ms | 64,68 |
| This work | DC: 76 kV/W, AC: 66.3 MHz/W | 5 | 0.01–0.22 | 308 μs | |

$^a$The VO$_2$-based oscillator of ref. 74 works in the visible range, and the central frequency of oscillations is within the range of MHz, so between 2 and 3 orders of magnitude higher than the carrier frequency of our setup. In this case, the frequency resolution or the normalized responsivity to the carrier frequency should be considered.

are compared to other works that report power sensing based on graphene, Silicon FETs, and some other VO$_2$-based devices as summarized in Table 1.

The idea of utilizing nonlinearity for detection is also explored in tunneling devices where steep $I$–$V$ curves result in strong rectification of incident electromagnetic fields[56,88,89].

Our results suggest that for the design and performance optimization of a spiking VO$_2$ sensor, it is important to: (i) design low-voltage actuation of structures (similarly with low-voltage-actuated MEMS with soft springs), therefore further miniaturization of device length and field concentrators are expected to help further enhancement of the sensing characteristics, (ii) improve the high-frequency power coupling into the 2-terminal VO$_2$ device by better antennas and CPW design and reduce the parasitic effects, (iii) consider operating the device near the switching point and combine with the use of a local heater.

In general VO$_2$ is a strong candidate for any type of applications requiring tunable materials. They provide an unprecedented ability for electromagnetic wave manipulation, particularly at THz frequencies. As for the particular detector reported in this work, a few applications are: (i) absolute THz power meters - for which our solution can provide a simpler and more cost-effective solution, (ii) sensing THz waves in a chip-scale technology that are currently emerging and for which the exact specifications are more difficult to be articulated, but can take benefit from or on-chip scalable VO$_2$ devices that can support building arrays of detectors.

## Methods

**VO$_2$ deposition**. During the development of pulsed laser deposition (PLD) VO$_2$ films at EPFL center of micro/nano-engineering clean room, we have selected the set of parameters (oxygen flow, pressure and temperature of annealing) that were producing the highest $R_{On}/R_{Off}$ and most reproducible results. The thin film deposition is done on a 525 μm-thick high resistivity float zone (HRFZ) Si wafer (>10,000 Ω cm). The substrate was then passivated by growth of 2 μm SiO$_2$ thermal wet oxide. The 100-nm-thick polycrystalline VO$_2$ film was synthesized on this substrate through a PLD system using a V$_2$O$_5$ target, in high-vaccum conditions, in a chamber with pressure of 0.01 mbar and Oxygen flow of 1 sccm. The laser energy was 400 mJ with a pulse frequency of 20 Hz. The substrate temperature was kept at 400 °C during the deposition, and the process was followed by post annealing at 475 °C. We have also controlled the purity of the VO$_2$ films and systematically checked no metal contamination in our samples. In this process, one achieves polycrystalline VO$_2$. However, there are other processes leading to mono-oriented VO$_2$ film deposition. For instance, monocrystalline Al$_2$O$_3$, which has a relatively low lattice parameter mismatch (4.5%) compared to VO$_2$ monoclinic phase. The polycrystalline film on top of Si/SiO$_2$ shows an overall lower On/Off ratio of conductivity with respect to that of a monocrystalline[16,17].

**Device fabrication**. The VO$_2$ film was patterned on a 100-nm-thick SiO$_2$ layer, using photolithography on negative tone AZ nLof 2020 resist followed by wet etching in diluted Cr etch solution. For the second layer, 15 nm Cr was sputtered on the patterned substrate as an interface layer and the process was followed by the sputtering of 150 nm Au. The connections and the 2-terminal devices were

completed by dry etching of the sputtered metal using a broad beam of Argon ions in an ion-beam-etcher. A more detailed process flow is presented in the Supplementary materials note 8: Fabrication Process.

**Electrical measurements**. High and low-frequency measurements were all performed on a Cascade Summit probe station. Mm-wave measurement incorporated broadband VNA system with MPI GSG 220-GHz infinity probes, each of them connected to a port from ANRITSU Vector Star VNA through frequency multiplier modules based on nonlinear transmission line (NLTL).

DC measurements were planned on a multicycle mode using Keithley semiconductor parameter analyzer. Each series of the DC measurements were performed at least a hundred cycles in a row to avoid transient results for the static characteristics of VO$_2$.

*Antennas*. According to Fig. 2d, the $|S_{21}|$ between the two ports is about −11 dB at its highest. We consider it when estimating the absorbed power by the VO$_2$-based antenna. Power loss also takes place through the cables, the transition from the infinity probes to the transmission line, the transition from the line to the dipole antenna, the return loss, and the reciprocal form of the mentioned transitions on the way to the receiving port.

According to the specifications in the frequency we use, the upper bound for the $|S_{21}|$ of the cables (Rosenberger RU5-08S1-08K1-00915 and TOTOKU TCF119XY100S2p) is collectively about −4 dB and the upper bound for that of an infinity probe is about −0.7 dB. We only estimated the lower band of losses on the results section, assuming that the mentioned ones here are addressed through the calibration.

## Data availability

All the relevant data supporting the results of this study are available from the corresponding authors upon request.

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

## Acknowledgements

This project has been funded by the European Research Council (ERC) via the Advanced ERC Grant Millitech, having as Principal Investigator professor Mihai Adrian Ionescu. The funder is the European Commission, grant agreement 695459.

## Author contributions

A.M.I. and F.Q. developed the device principle. F.Q. optimized the VO$_2$ thin film deposition recipe and fabricated the devices. F.Q. conducted static measurements on the sensor. F.Q. and T.R. worked on the oscillator circuit experiment. D.F. analyzed the bimodal distributions and the Allan deviation. F.Q., M.B. and J.L. conducted the high-frequency measurements. F.Q. and A.M.I. wrote the manuscript.

## Competing interests

The authors declare no competing interests.
