## [Peer Review File · Communications Materials]

3rd Aug 22

Dear Ms Qaderi,

Thank you for submitting your manuscript, "Millimeter-wave to near-terahertz sensors based on reversible insulator-to-metal transition in vanadium dioxide", to Communications Materials. It has now been seen by 3 referees, whose comments are appended below. You will see that while they find your work of potential interest, they have raised several major concerns that must be addressed. They have also made several suggestions to improve how the results are presented. In light of these comments, we cannot accept the manuscript for publication, but are interested in considering a revised version that addresses these serious concerns.

We hope you will find the referees' comments useful as you decide how to proceed. Should you be able to address these criticisms, we would be happy to look at a substantially revised manuscript. However, please bear in mind that we will be reluctant to approach the referees again in the absence of major revisions. If the revision process takes significantly longer than three months, we will be happy to reconsider your paper at a later date, as long as nothing similar has been accepted for publication at Communications Materials or published elsewhere in the meantime.

When submitting your revised manuscript, please include the following:

-A response letter with a point-by-point reply to each of the referee comments and a description of changes made. Please include the complete referee report in the response letter. Please note that the response letter must be separate to the cover letter to the editors.

-A marked-up version of the manuscript with all changes to the text in a different colored font. Please do not include tracked changes or comments. Please select the file type 'Revised Manuscript - Marked Up' when uploading the manuscript file to our online system.

-A clean version of the manuscript. Please select the file type 'Article File'.

-An updated <https://www.nature.com/documents/nr-editorial-policy-checklist.zip> Editorial Policy checklist, uploaded as a 'Related Manuscript File' type. This checklist is to ensure your paper complies with all relevant editorial policies. If needed, please revise your manuscript in response to these points. Please note that this form is a dynamic 'smart pdf' and must therefore be downloaded and completed in Adobe Reader. Clicking this link will download a zip file containing the pdf.

Please use the following link to submit your revised manuscript files:

[link redacted]

We understand that due to the current global situation, the time required for revision may be longer than usual. We would appreciate it if you could keep us informed about an estimated timescale for resubmission, to facilitate our planning. Of course, if you are unable to estimate, we are happy to accommodate necessary extensions nevertheless.

Please do not hesitate to contact me if you have any questions or would like to discuss the required revisions further. Thank you for the opportunity to review your work.

Best regards,

Klaas-Jan Tielrooij, PhD
Editorial Board Member
Communications Materials
orcid.org/0000-0002-0055-6231

Reviewers' comments:

Reviewer #1 (Remarks to the Author):

Review of the "Millimeter-wave to near-terahertz sensors based on reversible insulator-to-metal transition in vanadium dioxide"

The authors report a protocol enabling mm-wave to sub-THz radiation detection using strongly correlated compound VO₂ featuring field-stimulated metal-to-insulator transition at elevated temperatures. The approach is based on the shift of a critical voltage (i.e. voltage at which the transition occurs) in response to incident radiation. The devices of two types were studied: a) equipped with an antenna or b) integrated into the coplanar waveguides. Both types feature notable changes in the critical voltage upon exposing them to external radiation that is the main observable of the work. When normalized to the radiation power, this "photovoltage" translates to an exceptionally large responsivity reaching $R_v > 70$ kV/W. Moreover, the noise equivalent power (NEP), the key characteristic of radiation detectors, reported by the authors is of the order of $nV/Hz^{0.5}$ close to the room temperature that, in the sub-THz domain, is comparable to that of the best detectors available to date. The idea to explore MIT in VO₂ for mm-wave detection is, to the best of my knowledge, novel and interesting. However, considering the very large values of R_v and NEP, claimed by the authors, this draft can be published in Communications Materials only if the authors address the following concerns/questions/suggestions.

#1. The authors should provide a more detailed and accurate description of how NEP was measured. For example, the Johnson-Nyquist noise spectral density depends on the device

resistance. The authors should report this value. VO₂ exhibits MIT and thus it is not clear which resistance (metallic or insulating state) was taken. If the authors used the resistance of the metallic state, then it is not surprising that the NEP is low, but the device is not functioning as a field-effect transistor with metallic conductivity and thus this approach is wrong.

If the authors, on the contrary, measured the noise spectral density, they have to provide an accurate description of the measurement protocol. The key point of this paper is high responsivity and low NEP enabled by a novel detection principle. Therefore, an accurate description of the NEP and the NSD must be provided along with the data.

#2. Given that the responsivity drops drastically with increasing power, the authors should report for which power NEP in Fig. 6b is presented. The abstract and the main text should also emphasize that these high responsivity values were obtained for very small power levels (0.001 mW).

#3. The idea behind sensitive sub-THz detection in this work is similar to that of tunneling devices where steep I-V curves result in strong rectification of incident electromagnetic fields: Strong nonlinearities lead to efficient rectification. In this regard, the manuscript would benefit if the authors cite previous works on this subject:

Davids, P. S. et al. Infrared rectification in a nanoantenna-coupled metal-oxide-semiconductor tunnel diode. *Nat. Nanotechnol.* 10, 1033–1038 (2015).

Gayduchenko, I. et al. Tunnel field-effect transistors for sensitive terahertz detection, *Nat. Com.* 12:543 (2021).

Sharma, A., Singh, V., Bougher, T. L. & Cola, B. A. A carbon nanotube optical rectenna. *Nat. Nanotechnol.* 10, 1027–1032 (2015).

#4. In the abstract, the authors report responsivity values first in kHz/mW and then in kV/W. This is confusing. The same applies to the NEP which is typically expressed in W/sqrt(Hz), whereas in the abstract, the authors use nW. Furthermore, the authors use notations V_{DD} and V_{ds} and never define them in the main text; only in the Figure caption.

#5. Last but not least, I suggest the authors to amend the draft structure. The manuscript, in its present form, is repetitive, overworked, and is hard to read. For example, the last 5 paragraphs of the introductory section can be compressed into one short summarizing paragraph. A substantial portion of information (related to measurements and fabrication details) must be moved to the methods section. The section “VO₂ actuation through coupled antennas” and similar can be significantly compressed or moved to the Methods or Supplementary. In addition, the authors should avoid using phrases like “for the first time” or phrases like “which can be easily scaled” and “while the footprint is capable of being scaled down to a few hundreds of nanometers” in the abstract. It is not done in this work, why mention it? In addition, the supplementary file is corrupt and missing accurate figure referencing (see “??” marks in the text).

Reviewer #2 (Remarks to the Author):

The Manuscript by Qaderi et al. shows the realization and room temperature characterization of VO₂-based mm-wave detectors operating up to 220 GHz. The Topic of the manuscript is relevant and it can be of interest for the THz community. The experimental methods are explained with clarity. The idea of the astable circuit to exploit the IMT is particularly interesting and is the most important aspect of the work.

I think that the main weakness of the paper is a missing/inaccurate performance benchmark with respect to other technologies, in particular in terms of detector response time.

Moreover, the paper presents some language/presentation issues that make some aspects obscure and difficult to understand. These must be addressed before publication.

In the following, I will give a point-to-point list of comments to be addressed.

Abstract:

1. "low noise equivalent power (NEP) of 20nW at room temperature" The NEP should be expressed in units of power/Hz^{0.5}. This NEP is different to the value reported in Table 1.
2. The authors refer to "record" performance and responsivity "3 times better" than the state of the art. I think this is true only if one limits the comparison to other VO₂ devices, and not to the SOTA of room temperature THz detectors. Please consider revising this claim giving a clearer context.
3. "the footprint is capable of being scaled down to few hundreds of nanometers" This is not true for antenna-coupled architectures, because the footprint is limited by the size of the antenna itself.

Main:

4. The introduction section is mainly focused on VO₂-based detectors, but the benchmarking with respect to room temperature THz technologies is missing (commercially available systems, or proof-of-concept). I recommend the authors to introduce the state of the art of room temperature detectors, at least in the investigated frequency range (< 300 GHz).
5. Page 2: "The response time of our proposed sensor system is about 308us". The response time is a very important figure of merit for a photodetector. However, this figure of merit is not benchmarked with respect to other technologies (see Table 1). Moreover, by reading the main text, I don't understand how this value (308us) is experimentally measured and how the measurement setup is dictating this limit. I suggest moving the description of the measurement of the response time from the supporting information to the main text. The authors should include a detailed explanation about this. Would the usable response time be hampered by hysteresis in the experimental configurations of figure 1a, and 1b?
6. Page 2: "In bolometers, the lattice instabilities close to the phase-change temperature along with the local joule heating can interfere the bias control,". This is a bit too general, for which kind of bolometers is this statement valid? Moreover, I would rephrase with "... can interfere with the control bias".
7. Page 2: throughout the paper, the authors refer to the performance of their devices as "ultra-sensitive", "super-sensitive". This is not quantitative, and the proposed technology is not more sensitive than, e.g., commercially available Schottky diodes operating in the same frequency range (see www.vadiodes.com). I recommend avoiding this kind of claims.
8. Page 2: "The responsivity is at least 3 times larger than the similar sensors***, along with a limit of detection (LoD) comparable to the state of the art, and a footprint of one order of

magnitude smaller than the reported ones.” ***Please provide a reference. When referring to the state of the art, clarify that this comparison is only focused on VO₂ devices. Otherwise, rephrase or remove this sentence.

9. Page 2: “while further scaling the antennas design can shift the operational frequency up to THz.” Is this scaling straightforward? Is the effect broadband or is there an intrinsic cutoff frequency in the detection mechanism? Is the frequency range of operation solely determined by the antenna? Please explain and/or add references. Moreover, the authors should specify the geometry of the employed (and simulated) antennas: length, width, thickness.

10. Page 3: “To maintain the experiments reproducible, in all the configurations a heater (as a temperature reservoir) is connected to the chuck to keep the sample at a fixed temperature.” Can the author include a picture of the experimental setup? What is the “chuck”? Is it the substrate? How is the temperature controlled?

11. Page 3: “We exploit the collective effect of several excitation sources, here as DC bias voltage along with the mm-wave signal.” I understand that there are two excitations acting simultaneously, the DC bias and the RF signal. However, this sentence is quite confusing, why are there “several” sources? Can the authors rephrase or explain better?

12. Page 3: “high frequency...” and “...very low data rate”. Please quantify.

13. Figure 2(e) and 2(f) are not mentioned in the text and not explained.

14. Figure 3(b): what is the meaning of the shaded thickness around the solid blue and red lines? Is it the error? How is it evaluated (e.g. standard deviation, confidence interval)? Please define this in the corresponding figure legend, or in the figure caption.

15. Page 5: “...showing a noticeable bimodal behavior.” The authors refer to bimodal behaviour, but looking at Figure 4(d) through 4(f) it is difficult to understand the use of the term “bimodal”.

16. Page 5: I suggest changing the title of the section from “Readout circuit” to “Astable circuit”, this would better highlight the idea of measuring the oscillation frequency.

17. Page 6: “Hence, we report an upper bound for the NEP in this work.” What is reported is actually the external NEP, which is defined as the ratio between the noise figure and the external responsivity. In this sentence the authors are comparing the “external” NEP (eNEP) with the “internal” NEP (iNEP) (see Castilla et al. Nano Letters 2019, 19, 2765–2773 or Belacel et al. Nature Communications 8, 1578, 2017 for examples of internal vs. external responsivity calculations). Of course, the inequality $eNEP > iNEP$ is always fulfilled, however, it is confusing to declare that what is calculated in the manuscript is an upper bound of the NEP, it is the external NEP.

18. Page 6: I would suggest using the title “Response time” instead of “Dynamics”.

19. Page 7: “The cut-off frequency of the material switching suggests a timescale of 50 ps for a complete cycle of IMT and MIT together.” How is this sentence supported in the manuscript? Can the authors provide any evidence of this, or, at least, a reference?

20. Page 7: “The experimental figures of merit of the power sensor proposed here are comparable to or even outperform other works that report power sensing based on graphene, Silicon FETs, and some other VO₂-based devices as summarized in Table 1.” This sentence is wrong. By looking at Table 1, the proposed device has higher NEP with respect to graphene based and Silicon based technologies. Please note that the NEP is more important than the responsivity because it accounts for the detector noise level. Please note that the Response time is also an important figure of merit that gives, together with the NEP, the full picture of the performance of a single-pixel intensity detector. The response time is not

benchmarked in Table 1, the authors should include this figure of merit. Moreover, some additional comments on Table 1. The frequency range of Graphene-based devices is wrong: Ref. 54 reports detection at 2.8 THz and detection above 3 THz with NEP $<0.1 \text{ nW/Hz}^{0.5}$ is reported in Castilla et al. Nano Letters 2019, 19, 2765–2773. The frequency range of Si FET is wrong: detection above 4 THz has been reported in Lissauskas et al. J. Infrared Millim. Terahertz Waves 2014, 35, 63–80 (for a review on the topic see Javadi et al. Sensors 2021, 21(9), 2909). Table 1 should probably include other commercially available technologies, e.g. Schottky diodes and microbolometers, which actually have better performance (lower NEP, lower response time) with respect to the proposed architecture.

The paper has several typos. Please find below a non-exhaustive list.

Typos

Abstract:

- a. The acronym IMT is defined on its second occurrence, please revise.
- b. “State of the art. While...”

Main:

- c. Define the acronym IMT on its first occurrence
- d. Check that the “2” in “VO₂” is subscript in all the occurrences.
- e. Page 2: Define the acronym CPW in the main text.
- f. Page 3: “145 GHz as in return”, please rephrase (e.g. “145 GHz as shown in the return loss”).
- g. Page 4: replace “afford” with “trigger”.
- h. Page 4: “At bias voltages close to the IMT, the equivalent RMS AC voltage amplitude causing IMT is considerably smaller than the remaining DC voltage required for the IMT. Reciprocally, the shift in the IMT threshold voltage in presence of a high-frequency excitation is larger than the equivalent RMS AC voltage in the excitation.” Please consider rephrasing this sentence, it’s quite obscure.
- i. Define the acronyms RMS and MEMS in the text.
- j. Page 5: “having VO₂ half way of transition and above that.” Please consider rephrasing this sentence.
- k. Page 6: “..., is a criteria for the “response” of the...”, should be replaced by ““..., is a criterion for the response of the...”.
- l. Page 6: “The upper bound of the NEP in this case is near 5nW/Hz for bandwidths higher than 1Hz.” Do the authors mean “for frequencies above 1 Hz”?
- m. Page 6: “which is 2.77 mV when no power applied, implies the LoD.” What does “implies” mean here? Maybe “determines” or “defines” would fit better?
- n. Page 6: “acquisition”, “dominant”, please check the typos.

References

- o. Ref. 41 is missing a page and volume number
- p. Ref. 54 and ref. 56 are the same
- q. The authors of Ref. 57 are “Rama Murali G K, Prathmesh Deshmukh, S. S. Prabhu, and Palash Kumar Basu”
- r. In ref. 60: “Volterra” is upper-case.
- s. The frequency units should also be upper-case (GHz, THz).

Supporting

- t. Page 1: “ambience” should be replaced by “ambient”
- u. Page 2: some figure numbers are not specified.
- v. Figures 4 and 5 in SI are shown two pages after their supplementary notes (2 and 3, respectively). I suggest moving them, or separating the supplementary notes on different pages for more readability.

Reviewer #3 (Remarks to the Author):

The authors proposed and demonstrated 40-220 GHz sensors based on reversible insulator-to-metal transition in vanadium dioxide. The authors propose an electromagnetic wave-sensitive voltage-controlled spike generator based on the VO₂ switches in an astable circuit. The fabricated sensors show record figures of merit, such as responsivities of around 66.3 kHz/mW with a noise equivalent power (NEP) of 20 nW at room temperature, for a footprint of $2.5 \times 10^{-5} \text{ mm}^2$, which can be easily scaled. The authors claim that the solution gives 3 times better responsivity with only 1/10 footprint of the state of the art. I think the results are interesting, but several questions must be answered before this manuscript can be published.

Major points:

- 1- The authors stated in page 2 “The responsivity is at least 3 times larger than the similar sensors, along with a limit of detection (LoD) comparable to the state of the art”, the state-of-the-art reference was not referred here directly where here no absolute value is mentioned at which, frequency, NEP, temperature, and voltage bias. Can the authors explain more here?
- 2- 3 times better than the state of the art, it is still not much! maybe it is in the fabrication and measurement tolerances.
- 3- The authors didn't explore very important aspect related to the degradation and suppression of the MIT hysteresis loops in the electrical conductivity and optical reflectivity of amorphous VO₂ thin films after thermal cycles across the phase transition region. Or the reliability and lifetime of electrical switching based on the MIT in VO₂ thin films. Can the authors comment on that?
- 4- Which kind of THz application could this detector fit, taking in account the respectively, NEP, and reliability compared to other state-of-the-art THz detectors?
- 5- Is there any estimation about the detector performance, in terms of respectively, NEP, and bandwidth, in the case of coupling to an antenna not to CPW-based measurement?
- 6- In the methods section, (in VO₂ deposition part), the authors explained that 100-nm-thick polycrystalline VO₂ film was synthesized using a pulsed laser deposition (PLD) system. The same method (system) was used to synthesize single crystalline VO₂ film as explained here [1-3]. What is the direct effect of the VO₂ synthesis method on the device performance, stability, and reliability in case of single crystalline or polycrystalline VO₂? Generally, the

methods section was very short, I think the authors can explain much more especially in VO₂ deposition part.

[1] Dumas-Bouchiat F, Champeaux C, Catherinot A, Crunteanu A and Blondy P 2007 Appl. Phys. Lett. 91 223505

[2] Dumas-Bouchiat F, Champeaux C, Catherinot A, Givernaud J, Crunteanu A and Blondy P 2009 Materials and Devices for Smart Systems III ed J Su, L-P Wang, Y Furuya S Trolier-McKinstry and J Leng (Mater. Res. Soc. Symp. Proc.) 1129 275

[3] Aurelian, Crunteanu, et al. "Exploiting the semiconductor-metal phase transition of VO₂ materials: a novel direction towards tuneable devices and systems for RF-microwave applications." Advanced microwave and millimeter wave technologies semiconductor devices circuits and systems (2010): 35-56.

Minor points:

- 1- In the abstract, (IMT) abbreviations was mentioned in the 3rd line before the sentence in the 4th line where the explanation of the IMT came "insulator-to-metal transition (IMT)". Normally, acronyms (IMT) should be explained at the first time in the text not afterwards.
- 2- In Fig.3 b, the y-axis no name and no unit.

Response letter

Journal: *Nature Communications Materials*

Manuscript: COMMSMAT-22-0164

“Millimeter-wave to near-terahertz sensors based on reversible insulator-to-metal transition in vanadium dioxide”

Regarding the comments of the respected reviewer #1:

1. The authors should provide a more detailed and accurate description of how NEP was measured. For example, the Johnson-Nyquist noise spectral density depends on the device resistance. The authors should report this value. VO₂ exhibits MIT and thus it is not clear which resistance (metallic or insulating state) was taken. If the authors used the resistance of the metallic state, then it is not surprising that the NEP is low, but the device is not functioning as a field-effect transistor with metallic conductivity and thus this approach is wrong.

If the authors, on the contrary, measured the noise spectral density, they have to provide an accurate description of the measurement protocol. The key point of this paper is high responsivity and low NEP enabled by a novel detection principle. Therefore, an accurate description of the NEP and the NSD must be provided along with the data.

Answer:

We greatly appreciate the valuable suggestions and the constructive comments.

The Johnson-Nyquist noise in this device has been evaluated according to the insulating phase, where the resistance of the device is maximum 1.4 kΩ. So the upper bound for the thermal noise spectral density would be estimated as 4.8 nV·Hz^{-1/2} as a first order estimation of the noise spectral density.

To be more inclusive about other sources of noise such as material drifts, thermal fluctuations, flicker noise, and the noise implied by the setup. Of the static measurement, we applied a constant voltage and monitored the current fluctuations over time (200 s). After extracting the product of the current fluctuations and the voltage, we calculate the FFT of the time-domain signal. We normalize the measured values by the resolution bandwidth of the parameter analyzer (250 Hz). Then multiply the normalized power with the frequency difference between two adjacent measurement points to calculate the integral power between the two measurement points: (Therefore, summing up the calculated integral power per frequency step up to the first desired measurement bandwidth, the calculated value represents an integrated output noise power).

The NSD (v_N) could be calculated by:

$$v_N = \sqrt{P_{N,out} \cdot R_L}$$

Where $P_{N,out}$ is the output noise power spectrum and R_L is the measurement tool impedance, typically 50 Ω. In order to get the equivalent input noise power, we divide the v_N by the voltage responsivity value (R_V):

$$P_{N,in} = v_N / R_V$$

It is common to use the maximum value of R_V . This value is used in the curves of fig. 6 (b). Finally, to achieve the NEP we divide this power by the \sqrt{BW} . For the case of static detection, a value of 1 Hz is set as the BW.

We added a brief part of these explanations to the main text and the rest to the supplementary materials.

2. Given that the responsivity drops drastically with increasing power, the authors should report for which power NEP in Fig. 6b is presented. The abstract and the main text should also emphasize that these high responsivity values were obtained for very small power levels (0.001 mW).

Answer:

While calculating the NEP, it is common to use the maximum value of the responsivity [R1]. Moreover, the NEP is particularly important when the signal level leads to an SNR = 1, in other words the minimum detectable signal. Therefore, we used the responsivity at the minimum power that we measured.

We have now clarified and emphasized the small power levels while mentioning this responsivity in the abstract, as well as in the 4th paragraph under "detection capability", and in the caption of fig. 6 (b).

We also re-wrote and summarized the abstract again to fit in the 200-words rule of the journal.

[R1] Mackowiak V, Peupelmann J, Ma Y, Gorges A. NEP—noise equivalent power. Thorlabs, Inc. 2015;56.

3. The idea behind sensitive sub-THz detection in this work is similar to that of tunneling devices where steep I-V curves result in strong rectification of incident electromagnetic fields: Strong nonlinearities lead to efficient rectification. In this regard, the manuscript would benefit if the authors cite previous works on this subject:

Davids, P. S. et al. Infrared rectification in a nanoantenna-coupled metal-oxide-semiconductor tunnel diode. *Nat. Nanotechnol.* 10, 1033–1038 (2015).

Gayduchenko, I. et al. Tunnel field-effect transistors for sensitive terahertz detection, *Nat. Com.* 12:543 (2021).

Sharma, A., Singh, V., Bougher, T. L. & Cola, B. A. A carbon nanotube optical rectenna. *Nat. Nanotechnol.* 10, 1027–1032 (2015).

Answer:

We agree. Many thanks for the suggestion of the references. We added them to the manuscript.

4. In the abstract, the authors report responsivity values first in kHz/mW and then in kV/W. This is confusing. The same applies to the NEP which is typically expressed in W/sqrt(Hz), whereas in the abstract, the authors use nW. Furthermore, the authors use notations V_{DD} and V_{ds} and never define them in the main text; only in the Figure caption.

Answer:

We modified the text accordingly.

5. Last but not least, I suggest the authors to amend the draft structure. The manuscript, in its present form, is repetitive, overworked, and is hard to read. For example, the last 5 paragraphs of the introductory section can be compressed into one short summarizing paragraph. A substantial portion of information (related to measurements and fabrication details) must be moved to the methods section. The section “VO₂ actuation through coupled antennas“ and similar can be significantly compressed or moved to the Methods or Supplementary. In addition, the authors should avoid using phrases like “for the first time” or phrases like “which can be easily scaled” and “ while the footprint is capable of being scaled down to a few hundreds of nanometers” in the abstract. It is not done in this work, why mention it? In addition, the supplementary file is corrupt and missing accurate figure referencing (see “??” marks in the text).

Answer:

We have reconsidered the introductory paragraphs and re-wrote several last paragraphs. Also, some parts explaining the fabrication and measurement methods are moved to the methods/ supplementary. The mentioned phrases are removed from the text.

We have summarized and re-written the results section, especially for the most part of “VO₂ actuation through coupled antennas” and added some explanations to the methods section. Also revisited all the sections again as well as re-writing the abstract.

Regarding the comments of the respected reviewer #2:

We greatly appreciate the valuable suggestions and the constructive comments.

We updated the performance benchmarking with respect to other technologies in terms of detector response time.

1. Low noise equivalent power (NEP) of 20nW at room temperature” The NEP should be expressed in units of power/Hz^{0.5}. This NEP is different to the value reported in Table 1.

Answer:

We updated the table (5 nW/Hz^{0.5} is the correct value).

2. The authors refer to “record” performance and responsivity “3 times better” than the state of the art. I think this is true only if one limits the comparison to other VO₂ devices, and not to the SOTA of room temperature THz detectors. Please consider revising this claim giving a clearer context.

Answer:

We clarified this point in the abstract. We also re-wrote and summarized the whole abstract again to fit in the 200-words rule of the journal.

3. The footprint is capable of being scaled down to few hundreds of nanometers” This is not true for antenna-coupled architectures, because the footprint is limited by the size of the antenna itself.

Answer:

This sentence is regarding the active sensing area, which could be also incorporated in CPWs. We removed this sentence from the abstract for more clarity.

4. The introduction section is mainly focused on VO₂-based detectors, but the benchmarking with respect to room temperature THz technologies is missing (commercially available systems, or proof-of-concept). I recommend the authors to introduce the state of the art of room temperature detectors, at least in the investigated frequency range (< 300 GHz).

Answer:

Indeed, the reviewer is right, we have mainly focused on existing VO₂-based detectors in the introduction. Now, given the limited space of the paper and the purpose (this is not in itself a review paper) we have updated the introduction with a bit larger overview of the room temperature technologies state of the art and referred to some main players even though it was not the main scope goal of the paper to have a comprehensive review in the field.

We updated the introduction with an overview of the room temperature technologies state of the art main players and particularly, compared our sensor with other bolometric technologies for THz detection based on VO₂ in every aspect.

The added paragraph is highlighted in the introduction.

5. Page 2: “The response time of our proposed sensor system is about 308us”. The response time is a very important figure of merit for a photodetector. However, this figure of merit

is not benchmarked with respect to other technologies (see Table 1). Moreover, by reading the main text, I don't understand how this value (308us) is experimentally measured and how the measurement setup is dictating this limit. I suggest moving the description of the measurement of the response time from the supporting information to the main text. The authors should include a detailed explanation about this. Would the usable response time be hampered by hysteresis in the experimental configurations of figure 1a, and 1b?

Answer:

We explained the experimental origins of this time response in the main text under "Dynamics" subsection.

The time response of the oscillating sensor is mainly determined by the time constant of the astable circuit. The minimum frequency of oscillations for the proposed circuit is about 3.25 kHz which gives the maximum period of 308 us. We added the explanation to the part where we mention the response time.

6. Page 2: "In bolometers, the lattice instabilities close to the phase-change temperature along with the local joule heating can interfere the bias control,". This is a bit too general, for which kind of bolometers is this statement valid? Moreover, I would rephrase with "... can interfere with the control bias".

Answer:

Our comments refer to the operation of a VO₂ bolometer in the hysteresis region, or as it is called "transition edge". The reference we added to this sentence in the manuscript is about a bolometer operating at the transition temperature and the related challenges of operation in this region.

In a general review we have addressed how our method is different from bolometers, which is indeed a valid and important point on which we agree with the reviewer. We added references about the concept of bolometers in addition to the modification of the text for more clarity.

7. Page 2: throughout the paper, the authors refer to the performance of their devices as "ultra-sensitive", "super-sensitive". This is not quantitative, and the proposed technology is not more sensitive than, e.g., commercially available Schottky diodes operating in the same frequency range (see www.vadiodes.com). I recommend avoiding this kind of claims.

Answer:

We agree with the reviewer that, when compared to commercially available Schottky diodes, our devices may appear less sensitive. They are more sensitive than other emerging devices using VO₂ reported as research devices. For honesty and clarity, we removed these words from the parts describing our work.

8. Page 2: "The responsivity is at least 3 times larger than the similar sensors***, along with a limit of detection (LoD) comparable to the state of the art, and a footprint of one order of magnitude smaller than the reported ones." ***Please provide a reference. When referring to the state of the art, clarify that this comparison is only focused on VO₂ devices. Otherwise, rephrase or remove this sentence.

Answer:

We clarified this point and modified the text as the following:

"The responsivity is at least three times larger than the similar sensors [Ref1], along with a limit of detection (LoD) comparable to the state of the art [Ref2], and a footprint of one order of magnitude smaller than the typical of VO₂ sensors [Ref1]."

The added references are:

[Ref1] Kim, B.-J., Seo, G., Choi, J., Kim, H.-T. & Lee, Y. W. Laser-assisted control of electrical oscillation in VO₂ thin films grown by pulsed laser deposition. Jpn. J. Appl. Phys. 51, 107302 (2012).

[Ref2] Cosci, A. et al. Thz pyro-optical detector based on linbo₃ whispering gallery mode microdisc resonator. Sensors 17, 258 (2017).

9. Page 2: "while further scaling the antennas design can shift the operational frequency up to THz." Is this scaling straightforward? Is the effect broadband or is there an intrinsic cutoff frequency in the detection mechanism? Is the frequency range of operation solely determined by the antenna? Please explain and/or add references. Moreover, the authors should specify the geometry of the employed (and simulated) antennas: length, width, thickness.

Answer:

The design of the antennas in this work are limited according to the availability of measurement facilities such as the VNA and RF probes for much higher frequencies than reported in the work. We believe that the scaling is feasible through the same fabrication process for dimensions down to 1 μm . Therefore, the antenna scaling could be considered straight-forward within these dimensional limits.

In terms of the detection mechanism, the material shows responsivity to a wide range of input frequencies from RF to infrared, visible and even UV [Ref3-Ref5].

We added a brief explanation along with the references and antennas specifications to the paper as follows:

"According to the available facilities the antennas are designed for mm-wave frequency range, while further scaling the antennas design (which is feasible through the same fabrication process for dimensions down to 1 μm) can shift the operational frequency up to THz. In terms of the detection mechanism, the material shows responsivity to a wide range of input frequencies from RF to THz, infrared (IR), visible, and UV [Ref3-Ref5]."

[Ref3] Rosca, T. et al. "High Tuning Range Spiking 1R-1T VO₂ Voltage-Controlled Oscillator for Integrated RF and Optical Sensing." In ESSCIRC 2021-IEEE 47th European Solid State Circuits Conference (ESSCIRC), pp. 183-186. IEEE, 2021.

[Ref4] Chen, C. et al. "Optical phonons assisted infrared absorption in VO₂ based bolometer." Applied physics letters 91, no. 1 (2007): 011107.

[Ref5] Li, G. et al. "Photo-induced non-volatile VO₂ phase transition for neuromorphic ultraviolet sensors." Nature communications 13, no. 1 (2022): 1-9.

10. Page 3: "To maintain the experiments reproducible, in all the configurations a heater (as a temperature reservoir) is connected to the chuck to keep the sample at a fixed

temperature.” Can the author include a picture of the experimental setup? What is the “chuck”? Is it the substrate? How is the temperature controlled?

Answer:

The chuck is the metallic stage holding the substrate, as shown in fig. 7 (b) of the paper. It is connected to the temperature reservoir shown in Figure 1 here.

We added this information to the supplementary materials.

Figure 1. Temperature reservoir along with the setup

11. Page 3: “We exploit the collective effect of several excitation sources, here as DC bias voltage along with the mm-wave signal.” I understand that there are two excitations acting simultaneously, the DC bias and the RF signal. However, this sentence is quite confusing, why are there “several” sources? Can the authors rephrase or explain better?

Answer:

To avoid any further confusion, we rephrased it as:

"It is recognized that different excitation sources applied simultaneously can create a collective effect to assist the phase transition in correlated oxides such as VO₂."... "Here, we exploit DC bias voltage along with the mm-wave signal to influence the IMT/MIT transition points."

12. Page 3: “high frequency...” and “...very low data rate”. Please quantify.

Answer:

In the final corrected manuscript we have used the words "mm-wave" and "static" instead.

13. Figure 2(e) and 2(f) are not mentioned in the text and not explained.

Answer:

The citations of the two figures are now added into the manuscript. Thank you for the very careful check.

14. Figure 3(b): what is the meaning of the shaded thickness around the solid blue and red lines? Is it the error? How is it evaluated (e.g. standard deviation, confidence interval)? Please define this in the corresponding figure legend, or in the figure caption.

Answer:

It is indeed the standard deviation. A clarification has been added to the caption.

15. Page 5: "...showing a noticeable bimodal behavior." The authors refer to bimodal behaviour, but looking at Figure 4(d) through 4(f) it is difficult to understand the use of the term "bimodal".

Answer:

In most of the measurements, when we plot the distributions of these frequencies, they can be well fitted only with the superposition of two distributions that we called 'bi-modal'. We modified the part mentioning "bi-modal" and added the explanation.

The updated text:

"Most of the distributions cannot be fitted with normal Gaussian distribution curves and need a bi-modal fitting that could relate to nonlinear phenomena specific to stochastic systems like correlated oxides."

16. Page 5: I suggest changing the title of the section from "Readout circuit" to "Astable circuit", this would better highlight the idea of measuring the oscillation frequency.

Answer:

The change was implemented, thank you.

17. Page 6: "Hence, we report an upper bound for the NEP in this work." What is reported is actually the external NEP, which is defined as the ratio between the noise figure and the external responsivity. In this sentence the authors are comparing the "external" NEP (eNEP) with the "internal" NEP (iNEP) (see Castilla et al. Nano Letters 2019, 19, 2765–2773 or Belacel et al. Nature Communications 8, 1578, 2017 for examples of internal vs. external responsivity calculations). Of course, the inequality $eNEP > iNEP$ is always fulfilled, however, it is confusing to declare that what is calculated in the manuscript is an upper bound of the NEP, it is the external NEP.

Answer:

We agree and we clarified it as the "external" NEP.

18. Page 6: I would suggest using the title "Response time" instead of "Dynamics".

Answer: We agree and the change was implemented.

19. Page 7: "The cut-off frequency of the material switching suggests a timescale of 50 ps for a complete cycle of IMT and MIT together." How is this sentence supported in the manuscript? Can the authors provide any evidence of this, or, at least, a reference?

Answer:

Indeed, we realize that the reviewer is right as data to support the statement was missing. We added the following reference which includes our latest measurements on VO₂ switching for THz detection that supports the mentioned timescale:

Qaderi, F. et al. "Subthreshold VO₂ vertical switches for large-bandwidth millimeter-wave and sub-terahertz detection." In 2022 47th International Conference on Infrared, Millimeter and Terahertz Waves (IRMMW-THz), pp. 1-2. IEEE, 2022.

20. Page 7: "The experimental figures of merit of the power sensor proposed here are comparable to or even outperform other works that report power sensing based on graphene, Silicon FETs, and some other VO₂-based devices as summarized in Table 1." This sentence is wrong. By looking at Table 1, the proposed device has higher NEP with respect to graphene based and Silicon based technologies. Please note that the NEP is more important than the responsivity because it accounts for the detector noise level. Please note that the Response time is also an important figure of merit that gives, together with the NEP, the full picture of the performance of a single-pixel intensity detector. The response time is not benchmarked in Table 1, the authors should include this figure of merit. Moreover, some additional comments on Table 1. The frequency range of Graphene-based devices is wrong: Ref. 54 reports detection at 2.8 THz and detection above 3 THz with NEP <0.1 nW/Hz^{0.5} is reported in Castilla et al. Nano Letters 2019, 19, 2765–2773. The frequency range of Si FET is wrong: detection above 4 THz has been reported in c (for a review on the topic see Javadi et al. Sensors 2021, 21(9), 2909). Table 1 should probably include other commercially available technologies, e.g. Schottky diodes and microbolometers, which actually have better performance (lower NEP, lower response time) with respect to the proposed architecture.

Answer:

We modified the text about updated the benchmark with the response time and added more references on Schottky diodes and microbolometers.

However, the detector in "Castilla et al. Nano Letters 2019, 19, 2765–2773" is operating between 1.8 - 4.25 THz which is out of the frequency range we are operating, but for completeness, we added this one to the table too.

On the other hand, most of the publications reporting THz sensors do not report the response time, so the benchmark is not objectively covering the response time in each technology. To our knowledge, increasing the data rate may cause degradation in the responsivity, and when not reported, the characterization is generally done in static conditions.

21. Typos

We appreciate the precise reading and corrections by the respected reviewer. All the points are taken care of and highlighted in the manuscript (**we only listed here the ones with explanations. Others are just directly modified in the manuscript**).

- h. Page 4: "At bias voltages close to the IMT, the equivalent RMS AC voltage amplitude causing IMT is considerably smaller than the remaining DC voltage required for the IMT. Reciprocally, the shift in the IMT threshold voltage in presence of a high-frequency excitation is larger than the equivalent RMS AC voltage in the excitation." Please consider rephrasing this sentence, it's quite obscure.

Answer:

We rephrased it:

The equivalent RMS AC voltage amplitude corresponding to the mm-wave power that can trigger IMT in figs. 2(c) and 2(d) is smaller than the difference between the bias voltage and the threshold voltage. Reciprocally, the DC shift in the IMT threshold voltage of figs. 2(e) and 4(a) is greater than the equivalent RMS AC voltage amplitude of the corresponding mm-wave power.

i. Define the acronyms RMS and MEMS in the text.

Answer:

RMS is defined earlier in the 3rd paragraph under "detection capability".

We added the long version of MEMS (Micro-Electrical-Mechanical Systems).

j. Page 5: "having VO₂ half way of transition and above that." Please consider rephrasing this sentence.

Answer:

We rephrased it as: having VO₂ through semiconducting, mixed states and metallic state.

O, P, Q, R, S:

The errors had been caused by the citation bibtex files, now we modified them accordingly, thanks to the careful review.

T, U, V:

Done. Thanks again.

Regarding the comments of the respected reviewer #3:

1. The authors stated in page 2 “The responsivity is at least 3 times larger than the similar sensors, along with a limit of detection (LoD) comparable to the state of the art”, the state-of-the-art reference was not referred here directly where here no absolute value is mentioned at which, frequency, NEP, temperate, and voltage bias. Can the authors explain more here?

Answer:

We greatly appreciate all the valuable suggestions and the constructive comments.

References along with more details on operating point were added to this part of the manuscript.

The updated statement:

"The responsivity is at least three times larger than the similar sensors [Ref1], along with a limit of detection (LoD) comparable to the state of the art [Ref2], and a footprint of one order of magnitude smaller than the typical of VO₂ sensors [Ref1]."

The added references are:

[Ref1] Kim, B.-J., Seo, G., Choi, J., Kim, H.-T. & Lee, Y. W. Laser-assisted control of electrical oscillation in VO₂ thin films grown by pulsed laser deposition. Jpn. J. Appl. Phys. 51, 107302 (2012).

[Ref2] Cosci, A. et al. Thz pyro-optical detector based on linbo₃ whispering gallery mode microdisc resonator. Sensors 17, 258 (2017).

2. 3 time better than the state of the art, it is still not much! maybe it is in the fabrication and measurement tolerances.

Answer:

Indeed, one may consider that 3x is not much but we have systematic results that show that this is what can be honestly claimed with solid data. It could be possibly within the range of variations according to the measurements or fabrication. However, the prototyped sensors are not fully optimized for performance (e.g. with the device dimensions scaled, the film quality enhanced by PLD recipe optimization, insertion loss managed by substrate engineering and setup optimization and waveguide/ antenna design, etc.). So the value of "3 times" is what we can fairly report with our fabricated and measured proof of concept demonstrators as of the existing status.

We also re-wrote and summarized the whole abstract again to fit in the 200-words rule of the journal.

3. The authors didn't explore very important aspect related to the degradation and suppression of the MIT hysteresis loops in the electrical conductivity and optical reflectivity of amorphous VO₂ thin films after thermal cycles across the phase transition region. Or the reliability and lifetime of electrical switching based on the MIT in VO₂ thin films. Can the authors comment on that?

Answer:

The VO₂ film utilized in this work is poly-crystalline (as shown in the SEM image of fig. 2 (b) of the manuscript). As for the robustness and degradation, as important aspects in any future

applications, the reversible electric-field induced hysteresis loop has shown robustness over millions of non-stop switching cycles in our oscillator measurements. The same devices have been measured over days, showing the same V_{IMT} and V_{MIT} threshold voltages (of course, within their probabilistic distribution).

In case of keeping the devices safe from electrostatic discharge (ESD) in controlled environment (clean room), they have shown lifetimes of about 2 years since we fabricated them. Also, in case of passivating the VO_2 film surface, any degradation is negligible over months. However, most of our devices were not passivated and therefore a degradation in the ON/OFF ratio (between insulating and metallic states) has been observed when keeping them in ambient conditions. One can compare the switching ON/OFF ratio degradation over time according to the resistance characteristics of a VO_2 film depicted in Figure 2: Measurement 2 is taken about 2 months after Measurement 1 on the same VO_2 film, and Measurement 3 is taken after almost a year.

This is while the threshold voltages are slightly affected and therefore our method of sensing barely suffers from that. In the static approach (figs. 2(e) and 4(a)), the resistivity in either of the states is not even used in the detection method. In the oscillator approach, the resistivity could influence the time constants of the circuit, but again, we refer to the relative shift in the frequency of oscillations with respect to a baseline which is the free running with no external power exposed. So the sensor remains functional and only re-calibrations are needed.

We added these explanations to the supplementary materials and briefly mentioned robustness in the manuscript and referred to degradation (see below).

Figure 2. sample measurements of MIT switching degradation over time. Measurement 2 is taken about 2 months after Measurement 1 on the same VO_2 film, and Measurement 3 is taken after almost a year.

The text added to the supplementary materials:

"The reversible electric-field induced hysteresis loop has shown robustness over millions of non-stop switching cycles in our oscillator measurements. The same devices have been measured over days, showing the same V_{IMT} and V_{MIT} threshold voltages.

In case of keeping the devices safe from electrostatic discharge (ESD), they have shown lifetimes of about 2 years since we fabricated them. Also, in case of passivating the VO_2 film surface, the degradation is negligible over months. However, most of our devices were not passivated and therefore a degradation in the ON/OFF ratio (between insulating and metallic states) has been

observed. One can compare the switching ON/OFF ratio degradation over time according to the resistance characteristics of a VO₂ film depicted in fig.5: Measurement 2 is taken about 2 months after Measurement 1 on the same VO₂ film, and Measurement 3 is taken after almost a year.

This is while the threshold voltages are slightly affected and therefore our method of sensing barely suffers from that: In the static approach (figs. 2(e) and 4(a) of manuscript), the resistivity in either of the states is not even used in the detection method. In the oscillator approach, the resistivity could influence the time constants of the circuit, but again, we refer to the relative shift in the frequency of oscillations with respect to a baseline which is the free running with no external power exposed. So the sensor remains functional and only re-calibrations are needed."

4. Which kind of THz application could this detector fit, taking in account the respectively, NEP, and reliability compared to other state-of-the-art THz detectors?

Answer:

First, we can state that in general VO₂ is a strong candidate for any type of applications requiring tunable metamaterials provide an unprecedented ability for electromagnetic wave manipulation, particularly at THz frequencies. As for the particular detector reported in this work, there are a few applications: (i) *absolute Terahertz power/energy meters* – for which our solution can provide a simpler and more cost-effective solution while remaining within an accuracy better than 10% and NEP typically in the range of nW to $\mu\text{W}/\text{Hz}^{1/2}$, (ii) applications of sensing THz waves in a chip-scale technology that are currently emerging and for which the exact specifications and more difficult to be articulated but can take benefit from or on-chip scalable VO₂ devices that can support building arrays of detectors.

We mentioned it briefly at the end of the discussion section in the manuscript.

5. Is there any estimation about the detector performance, in terms of respectively, NEP, and bandwidth, in the case of coupling to an antenna not to CPW-based measurement?

Answer:

The antennas measurements were performed using Anritsu equipment at ETH Zürich available for a limited time. that the respective setup did not support a fully detailed oscillator experiment and NEP measurements, as the reviewer is suggesting. However, below we provide a discussion of the comparison of VO₂ devices coupled to the antennas versus the ones inside CPWs in the following terms:

- Responsivity:

The static responsivity for the VO₂ devices inside CPWs (fig. 4 in the manuscript) shows how the IMT threshold voltage (V_{IMT}) is affected by the absorption of mm-wave power. This responsivity is also measured for the antennas in fig. 2 (e) of the manuscript.

Analytically, one can calculate the frequency of oscillations in the astable circuit of fig. 1 (c) of the manuscript based on the VO₂ threshold voltages and other values of the circuit. Each cycle of the oscillations is formed of two time-constants, one by the ON state of the VO₂ device and the other by the OFF state of the material.

When VO₂ is in ON state, the rise time of the spike is determined by R_{ON} and the capacitance. The amount of time that the oscillator circuit spends at ON state in a single period is calculated by:

$$t_1 = -\frac{R_{ON}C}{\alpha_{ON}} \ln\left[\frac{V_{DD} - I_D R_{ON} - (V_{DD} - V_{MIT})\alpha_{ON}}{V_{DD} - I_D R_{ON} - (V_{DD} - V_{MIT})\alpha_{ON}}\right]$$

Where R_{ON} is the VO₂ resistance in metallic state, $\alpha_{ON} = 1 + R_{ON}/r_o$, and r_o is the output resistance of the transistor. And the amount of time that the oscillator circuit spends at OFF state in a single period is calculated by:

$$t_2 = -\frac{R_{OFF}C}{\alpha_{OFF}} \ln\left[\frac{V_{DD} - I_D R_{OFF} - (V_{DD} - V_{MIT})\alpha_{OFF}}{V_{DD} - I_D R_{OFF} - (V_{DD} - V_{MIT})\alpha_{OFF}}\right]$$

Where R_{OFF} is the VO₂ resistance in insulating state and $\alpha_{OFF} = 1 + R_{OFF}/r_o$.

Therefore, the period of oscillations is analytically determined by the threshold voltages. Using the data from fig. 2 (e) in the manuscript, one can calculate the AC responsivity of the antenna-coupled devices.

- Bandwidth:

The overall bandwidth of the detector based on the antenna-coupled devices is influenced by the readout circuit as well as the antenna's bandwidth. The latter is about 3 GHz, so it is not limiting the bandwidth imposed by the readout circuit, which is below 4 kHz in this design. However, the concern remains valid about the antennas limiting the bandwidth in case of using a faster readout circuit.

- NEP

We address the noise associated with the VO₂ device itself, through direct noise measurements on the two-terminal device (We have added more details on this measurement to the main text about NEP as well as the supplementary materials)

The NSD (v_N) could be calculated by:

$$v_N = \sqrt{P_{N,out} \cdot R_L}$$

Where $P_{N,out}$ is the output noise power spectrum (which we measured directly) and R_L is the measurement tool impedance, typically 50 Ω . In order to get the equivalent input noise power, we divide the v_N by the voltage responsivity value (R_V):

$$P_{N,in} = v_N/R_V$$

In the latter equation, we should replace the R_V with the one from the antennas. Thus, the value used in fig. 6 (b) of the manuscript should on the one hand get a scale of $R_{V,CPW}/R_{V,antennas}$. On the other hand, to achieve the NEP we divide this power by the \sqrt{BW} . In non-static measurements one should also consider the antenna's bandwidth influence, as it has a filtering effect on the input noise to the device and it can also condition the measurement bandwidth. Therefore, by two updated factors of responsivity and bandwidth for the antenna-coupled devices, one could estimate the new NEP for them.

6. In the methods section, (in VO₂ deposition part), the authors explained that 100-nm-thick polycrystalline VO₂ film was synthesized using a pulsed laser deposition (PLD) system. The same method (system) was used to synthesize single crystalline VO₂ film as explained here [1-3]. What is the direct effect of the VO₂ synthesize method on the device performance, stability, and reliability in case of single crystalline or

polycrystalline VO₂? Generally, the methods section was very short, I think the authors can explain much more especially in VO₂ deposition part.

Answer:

The effect of the PLD synthesis parameters of the polycrystalline VO₂ films on the device performance and reliability is in itself a very interesting yet complex and long study that is beyond the scope of this paper.

During the development of a PLD deposition at EPFL we have selected the set of parameters (oxygen flow, pressure and temperature of annealing) that were producing the highest R_{On}/R_{Off} and most reproducible results. The set of optimal parameters are reported in the paper as:

"The 100-nm-thick polycrystalline VO₂ film was synthesized on a HRFZ Si(500 um)/ SiO₂(2 um) substrate through a PLD system using a V 2 O 5 target, in high-vacuum conditions, in a chamber with pressure of 0.01 mbar and Oxygen flow of 1 sccm. The laser energy was 400 mJ with a pulse frequency of 20 Hz. The substrate temperature was kept at 400°C during the deposition, and the process was followed by post annealing at 475°C."

We have also controlled the purity of the VO₂ films and systematically checked no metal contaminations in our samples but we are not able to provide a fully technology optimization study within the scope of this paper.

The authors thank the reviewer for the suggested references regarding PLD synthesized VO₂. The RF/ MW applications of tunable structures based on various types of VO₂ layers in [Ref1] and [Ref3] were particularly of our interest and we referred to them in the introduction of our manuscript by adding the following sentence:

"Moreover, RF/ MW applications of tunable structures are reported based on various types of VO₂ layers [Ref1, Ref3]."

It is worth mentioning that in [Ref1] the VO₂ film is fabricated on the two types of substrate: Al₂O₃(C) and Si/SiO₂. According to [Ref1], since monocrystalline Al₂O₃ has a relatively low lattice parameter mismatch (4.5%) compared to VO₂ monoclinic phase, it is a good candidate to deposit mono-oriented VO₂ films. On the other hand, in the case of VO₂ on Si/SiO₂, the crystalline lattice mismatch is large, and a poly-crystallized film growth is observed. Hence, in the case of our work, one would expect the poly-crystalline structure according to the stack of Si/SiO₂ used in our devices.

Comparing the VO₂ samples on Si/SiO₂ and the Al₂O₃ substrate in these references, the poly-crystalline film on top of Si/SiO₂ shows an overall lower On/Off ratio of conductivity with respect to the one on a mono-crystalline.

We added the above points briefly to the methods, under "VO₂ deposition".

[Ref1] Dumas-Bouchiat F, Champeaux C, Catherinot A, Crunteanu A and Blondy P 2007 Appl. Phys. Lett. 91 223505

[Ref2] Dumas-Bouchiat F, Champeaux C, Catherinot A, Givernaud J, Crunteanu A and Blondy P 2009 Materials and Devices for Smart Systems III J Su, L-P Wang, Y Furuya S Trolier-McKinstry and J Leng (Mater. Res. Soc. Symp. Proc.) 1129 275

[Ref3] Aurelian, Crunteanu, et al. "Exploiting the semiconductor-metal phase transition of VO₂ materials: a novel direction towards tuneable devices and systems for RF-microwave applications."

Advanced microwave and millimeter wave technologies semiconductor devices circuits and systems (2010): 35-56.

18th Jan 23

Dear Ms Qaderi,

Your manuscript titled "Millimeter-wave to near-terahertz sensors based on reversible insulator-to-metal transition in vanadium dioxide" has now been seen again by our referees, whose comments appear below. In light of their advice I am delighted to say that we are happy, in principle, to publish a suitably revised version in Communications Materials under the open access CC BY license (Creative Commons Attribution v4.0 International License).

We therefore invite you to revise your paper one last time to comply with our journal policies and formatting style, in order to maximise the accessibility and therefore the impact of your work.

EDITORIAL REQUESTS

* Your manuscript should comply with our policies and format requirements, detailed in our style and formatting guide (<https://www.nature.com/documents/commsj-phys-style-formatting-guide-accept.pdf>).

* Please edit your manuscript according to the editorial requests in the attached table, and outline revisions made in the right hand column. If you have any questions or concerns about any of our requests, please do not hesitate to contact me. It is important that each request be addressed in order to avoid delays in accepting your manuscript. Please upload the completed table with your manuscript files as a Related Manuscript file.

* The editorial requests table also includes a full list of the files that must be provided upon resubmission. Please upload your files according to this table.

* An updated editorial policy checklist that verifies compliance with all required editorial policies must be completed and uploaded with the revised manuscript. All points on the policy checklist must be addressed; if needed, please revise your manuscript in response to these points. Please note that this form is a dynamic 'smart pdf' and must therefore be downloaded and completed in Adobe Reader. Clicking this link will download a zip file containing the pdf.

OPEN ACCESS

Communications Materials is a fully open access journal. Articles are made freely accessible on publication under a [CC BY](http://creativecommons.org/licenses/by/4.0) license (Creative Commons Attribution 4.0 International License). This license allows maximum dissemination and re-use of open access materials and is preferred by many research funding bodies.

For further information about article processing charges, open access funding, and advice and support from Nature Research, please visit <https://www.nature.com/commsmat/about/open-access>

RESUBMISSION

At acceptance, you will be provided with instructions for completing this CC BY license on behalf of

all authors. This grants us the necessary permissions to publish your paper. Additionally, you will be asked to declare that all required third party permissions have been obtained, and to provide billing information in order to pay the article-processing charge (APC).

Please use the following link to submit your revised files:

[link redacted]

We hope to hear from you within two weeks; please let us know if the process may take longer.

Best regards,

Klaas-Jan Tielrooij, PhD
Editorial Board Member
Communications Materials
orcid.org/0000-0002-0055-6231

&

Dr Aldo Isidori
Senior Editor
Communications Materials

REVIEWERS' COMMENTS:

Reviewer #1 (Remarks to the Author):

The authors addressed all my questions and I recommend the publication of the manuscript.

Reviewer #2 (Remarks to the Author):

I think the paper has been properly revised, and can be published.